# Hybrid Nanobioengineered Nanomaterial-Based Electrochemical Biosensors

**DOI:** 10.3390/molecules27123841

**Published:** 2022-06-15

**Authors:** Dayana Soto, Jahir Orozco

**Affiliations:** Max Planck Tandem Group in Nanobioengineering, Institute of Chemistry, Faculty of Natural and Exact Sciences, University of Antioquia, Complejo Ruta N, Calle 67 N° 52–20, Medellin 050010, Colombia; ingriddayana.soto@udea.edu.co

**Keywords:** hybrid, nanobiomaterial, bioreceptor, bioaffinity, biocatalytic, biosensor, cytosensor, genosensor, immunosensor, electrochemical

## Abstract

Nanoengineering biosensors have become more precise and sophisticated, raising the demand for highly sensitive architectures to monitor target analytes at extremely low concentrations often required, for example, for biomedical applications. We review recent advances in functional nanomaterials, mainly based on novel organic-inorganic hybrids with enhanced electro-physicochemical properties toward fulfilling this need. In this context, this review classifies some recently engineered organic-inorganic metallic-, silicon-, carbonaceous-, and polymeric-nanomaterials and describes their structural properties and features when incorporated into biosensing systems. It further shows the latest advances in ultrasensitive electrochemical biosensors engineered from such innovative nanomaterials highlighting their advantages concerning the concomitant constituents acting alone, fulfilling the gap from other reviews in the literature. Finally, it mentioned the limitations and opportunities of hybrid nanomaterials from the point of view of current nanotechnology and future considerations for advancing their use in enhanced electrochemical platforms.

## 1. Introduction

Nanobioengineered-based hybrid electrochemical biosensors exploit the synergistic properties of hybrid systems that connect biomolecules with nanomaterials to engineer highly sensitive biosensing platforms for the specific electrochemical detection of different target analytes. Nanobioengineered platform-based electrochemical biosensors have been implemented in biomedicine, environmental, food, and security industries, demonstrating their versatility and great potential. Notably, in the biomedical field, modern nanobioengineered biosensing devices are escalating their horizon to face multitudinous medical complications, i.e., providing early, accurate, and specific diagnoses of diseases [1,2,3]. 

Electrochemical nanobiosensors are devices designed as alternatives to conventional laboratory-based detection techniques for disease diagnosis due to their robustness, small size, user-friendly operation, amenability for miniaturization, and potential for personalized diagnosis [4,5,6]. Electrochemical nanobiosensors comprise electroactive transducer platforms that anchor specific and selective bioreceptors, generating a nanobioconjugate. The nanobioconjugate uses different interactions between enzymes, antibodies, DNA(RNA) strands, cell-organelles, proteins, peptides, glycans, etc., with target analytes to report superficial electrochemical changes [7]. The sensing mechanism involves the target analyte–bioreceptor interaction, generating a stimulus transduced (transformed) into a decipherable signal that correlates to the target analyte’s concentration in a particular sample [8]. 

Despite the tremendous promise of electrochemical nanobiosensors in biomolecular analysis, they sometimes suffer from poor sensitivity and short shelf life. The limit of detection (LOD) and narrow dynamic range often limit their practical application, hindering their path toward the market [9]. In this context, current investigations in the biosensing field aim to engineer nanomaterials that, coupled with biomolecules and transducer platforms, can give rise to specific and versatile nanobiohybrid-based biosensors to address the commented limitations, paving the way toward real solutions. 

In recent years, researchers have dedicated efforts to harnessing the unique atomic and molecular properties of nanobioengineered nanomaterials, including carbon nanomaterials, semiconductor/conductor polymers, metallic nanoparticles, and their nanoconjugates [10,11,12]. Such nanobiostructures may improve the interaction with the bioreceptors, thus dramatically amplifying the resultant signal, lowering the LOD, extending the linear detection range, shortening the testing time, and increasing the long-term stability of the detection systems [13,14,15].

Given such recent developments in nanomaterial engineering, this review reports understandings of advanced nanomaterials, giving an insight into the elucidation of their role in enhancing the analytical performance of electrochemical biosensors, unlike other reports from the literature. In this context, types of hybrid nanomaterials are systematically categorized, discussing their importance in detecting different target analytes of biomedical interest. Special attention is given to electrochemical biosensors incorporating various hybrid nanomaterials, explaining the chemistry of functionalization with biomolecules for assembling the biosensing systems and signal enhancement strategies’ fundaments, differencing elements concerning the state-of-the-art. Finally, a selected representative group of applications of nanomaterials in electrochemical biosensing were discussed to provide a more comprehensive background, including the detection of protein, peptidic, and genetic-disease-related biomarkers; small molecules; and metabolites among others. It is anticipated that this review will be helpful for newcomers in the field and show them the importance of nanobiohybrid-biosensors in real-time medical diagnostic applications published in the last years.

## 2. Nanohybrids and Nanocomposites

Nanostructured nanomaterials, both nanohybrids and nanocomposites, have been increasingly exploited in developing electrochemical biosensors [16] and functional interfaces [17,18,19,20,21] with enhanced properties in terms of sensitivity, selectivity, robustness, and simplicity [22]. 

Nanocomposite materials are prepared by combining two or more different materials with different physicochemical properties, where one of the constituents has dimensions at the nanoscale or, instead, the nanocomposite structure exhibits a nanometric phase separation of the individual components. In preparing nanocomposites, one of the constituent materials acts as a support matrix in which other materials called reinforcement agents are incorporated [23,24]. Nanocomposites present mixed properties based on the original properties of each constituent nanomaterial, not modified during the preparation process [25].

Similarly, a hybrid nanomaterial combines organic and inorganic building blocks [4], which present a continuous interface between the structural components [25], and new, improved physicochemical properties emerge that are distinct from the specific properties of the components alone [26]. Hybrid nanomaterials can function as novel electrode materials, signal amplifiers, and catalysts of the electrochemical reaction of the product generated in situ during the biorecognition event. To date, the most common hybrid nanomaterials applicable to electrochemical biosensing include metallic nanostructures [27,28], silicon nanomaterials [29,30,31,32], carbon nanostructures [17,33,34], and semiconductor polymers [35,36,37,38,39], with great potential for the development of electrochemical nanobiosensors with enhanced performance [40,41,42], as commented. This section will comment on the main examples of the last ten years (Table 1), focused on nanostructured nanomaterials employed in developing nanohybrids for their implementation in electrochemical biosensing.

### 2.1. Metallic Nanostructures

Metal nanostructures are generally defined as isolable particles between 1 and 50 nm in size, usually protected within shells to avoid agglomeration. Such nanoparticles have physical, electronic, and chemical properties different from bulk metals due to their small size [43]. Metallic nanostructures (MNS) (e.g., Au, Pt, Cu, Ir, Zn, Ag, Fe, Pd, and their different alloys) have been used in electrochemical biosensing devices to improve their conductive properties [29,33,44] as mediators of the signal from target-bioreceptor recognition [45] and to promote a better contact between bioreceptors and transducers, thus improving the sensitivity of the resultant devices. They have shown high surface area, chemical stability, and high compatibility and conductivity [44]. 

The properties of metal nanostructures strongly depend on the number and kind of atoms that make up the particle. Typically, such particles are prepared by chemical reduction of the corresponding transition metal salts in the presence of a stabilizer (capping agents such as citrate or thiols), which binds to their surface to impart high stability and rich linking chemistry, and provides the desired charge and solubility properties [46,47]. However, implementation of inorganic particles in biosensors is limited by changes in their oxidation state due to variations in conditions of the medium such as pH, ionic strength, and temperature upon time. 

Incorporating hybrid systems based on metallic nanostructures in transducer platforms has improved the properties of the resulting electrochemical biosensors concerning the concomitant constituent counterparts acting alone. Xue et al. constructed a three-dimensional (3D) architecture consisting of layer-by-layer graphene–gold nanorod (GNR) for amperometric detection of hydrogen peroxide (H_2_O_2_) and glucose [48]. The 3D hybrid graphene-GNR-modified electrode exhibited the highest sensitivity compared with the active carbon, graphene-oxide, cysteine-graphene oxide, and GNR-coated electrodes, demonstrating improved properties of hybrid nanostructures. The sensor was tested with different H_2_O_2_ concentrations having a linear response from 0 to 5.0 mM and a 2.9 µM LOD. In addition, the 3D layer-by-layer nanomaterial-modified-electrode was used to detect glucose, showing excellent sensitivity, selectivity, and minimum interference by ascorbic acid (AA) and uric acid (UA) in the test solution. Further, Hashemi et al. developed a retrievable electrochemical biosensor based on Ag and hybrid Ag-Fe_3_O_4_ metallic nanoparticles for precise, real-time, and repeatable detection of ascorbic acid (AA) within the blood plasma samples. The modified electrode with GO-Ag-Fe_3_O_4_ showed a LOD and sensitivity of 74 nM and 1146.8 μA/mM·cm^−2^, respectively, within the concentration range of 0.2–60 μM. Additionally, the modified electrode kept 91.23% of its total performance after 15 days of performance, highlighting its superior stability [49].

Metallic hybrid nanostructures can also be implemented to support the immobilization of different bioreceptors. Li et al. fabricated an electrochemical biosensor based on a Pt-Au nanowire/Au nanoparticle hybrid array for the simultaneous detection of penicillin and tetracycline by cyclic voltammetry (CV). The nanostructured hybrid consisted of vertically aligned Pt-Au nanowires electrodeposited within anodic aluminum oxide (AAO) membranes. The authors incorporated L-cysteine to form a monolayer in the Au segment as a bioreceptor for tetracycline detection. They coated the Au nanoparticles on the Pt nanowire segments to immobilize penicillinase on the Au NPs surface with EDC/NHS as a crosslinker. The prepared Au(L-cysteine)-Pt(penicillinase) nanowire array electrode showed the ability to detect penicillin and tetracycline simultaneously (Figure 1I), with a high sensitivity of 41.2 and 26.4 μA/(μM·cm^2^) for penicillin and tetracycline, respectively. The advantages of the hybrid nanowire/NP array structure make the electrochemical biosensor an enhanced platform for the simultaneous detection of various bioanalytes [50]. 

Metallic nanostructures have been incorporated into polymeric matrices to promote miniaturization and the high analytical performance of sensing platforms. Kadian et al. designed a nanohybrid film based on indium tin oxide/TiO_2_/poly(3-hexylthiophene) (PHT) for glucose detection by chronoamperometry. The thin film of hybrid PHT/TiO_2_ was deposited onto indium-tin-oxide (ITO) glass substrate, followed by immobilization of glucose oxidase (GOx). Under the optimized experimental conditions, the electrochemical biosensor showed a wide linear range between 1–310 mg/dL with a LOD of 0.62 mg/dL. The developed biosensor was applied to detect glucose in human saliva samples without any pretreatment step with a time response of less than 10 s, suggesting a miniature device for glucose detection at the point of care [53]. 

Paul et al. proposed a new glucose biosensor using TiO_2_ nanoparticles and glucose oxidase encapsulated into a ZIF-8 metal–organic framework (MOF) cavity, as shown in Figure 1II. The synthesized nanomaterial was drop-casted on a glassy carbon electrode and the sensor response was measured by amperometry [51]. Double-layer capacitance and high surface area of TiO_2_ increased catalytic activity of the biosensor to detect a low concentration level of 80 nM in a linear range of 2 to 10 mM.

Hybrid metallic nanostructures have been used as signal amplifiers in developing electrochemical biosensors. Tian et al. reported a planar nanohybrid of VS_2_/AuNP and CoFe_2_O_4_ nanozyme for signal amplification in the sensitive quantification of kanamycin (Kana), as shown in Figure 1III. The detection platform employed VS_2_/AuNPs as the support material due to their excellent conductivity, high specific surface area, and hybridized complementary Kana-biotinylated aptamer hairpin DNA (hDNA). In addition, streptavidin-functionalized CoFe_2_O_4_ nanozyme with higher peroxidase-like catalytic activity was immobilized on the aptasensor. The electrochemical signal was linear from 1 pM to 1 μM in Kana quantification with a LOD of 0.5 pM. Different metallic nanostructures assembled in the aptasensor led to excellent analytical performance and precision. Hence, it could detect various targets by replacing the corresponding aptamer with an on-demand specific one [52].

### 2.2. Silicon Nanomaterials

Recently, silicon materials have attracted increased attention in biosensing electrochemical platforms due to their physicochemical and structural properties. Silicon nanomaterials have high mechanical resistance, thermal stability, long functional life, and versatility. In addition, it is possible to obtain a wide range of porous textures, well-ordered network structures, and surface functionalities that favor incorporating various biorecognition elements for highly efficient sensing platforms [54,55,56]. In most cases, silicon materials can be modified or functionalized by wet chemistry and electrochemical methods. 

Silicon nanometals synthesized in the presence of surfactants as molding agents of the organized porous structure were reported for the first time by Mobil in 1992 [54]. These silicon nanometals have evolved in terms of complexity regarding the combination of porosity (hierarchical silicas) and the incorporation of organic entities in their structure (organosilicates) [57,58]. They have been grouped by families, such as the MCM-X (synthesized by the Mobil Corporation with cationic surfactants) and the SBA-X (from the University of Santa Barbara with triblock copolymer-type neutral surfactants such as Pluronic P123 and F127). The organosilicates can be obtained through one-step synthesis methods by co-condensation of the silica source such as sodium silicate, alkoxysilanes such as tetraethylorthosilicate (TEOS), among others, with alkylalkoxysilanes such as 3-aminopropyltriethoxysilane (APTES), phenyltrimethoxysilane (PTMS), phenyltriethoxysilane (PTES), among others, forming interconnected siliceous networks through organic bridges. Organosilicates can also be produced by postsynthesis processes, replacing the silanol groups of the synthesized silica with organic groups through silanization reactions with alkylalkoxysilanes having amino, epoxy, glyoxyl, aryl, phenyl, alkyl groups [30,59,60]. Hybrid nanomaterials based on silicon have been implemented in the immobilization of bioreceptors or amplification systems, since the presence of organic groups produces changes in the hydrophobic and electrochemical properties of the surface, which promote different bioreceptor–electrode surface interactions, generating a favorable microenvironment for the stability of the different bioreceptors in electrochemical biosensing approaches [60,61]. 

Zhang et al. designed a novel sensitive sandwich-type pseudoenzyme aptasensor to detect thrombin. The greatly enhanced sensitivity was based on a mesoporous silica multiwall carbon nanotube (mSiO_2_@MWCNT)-based nanohybrid as improved materials and a pseudobienzymic electrocatalytic system. mSiO_2_@MWCNT nanocomposites were biocompatible and had a suitable microenvironment to stabilize aptamer assembly and load large amounts of electron-mediating thionin (Thi), platinum nanoparticles (PtNPs), and hemin/G bioelectrocatalytic complex–quadruplex. Furthermore, in the presence of H_2_O_2_ in an electrolytic cell, the synergistic reaction of PtNPs and hemin/G-quadruplex bioelectrocatalyzed H_2_O_2_ reduction, thus dramatically amplifying Thi electron mediator response signals and improving sensitivity. Furthermore, dendrimer-functionalized reduced graphene oxide (PAMAM-rGO) as a biosensor platform enhanced the surface area for immobilization of abundant primary aptamers and facilitated electron transfer from Thi to the electrode, thus amplifying the sensing response. As a result, the aptasensor showed high sensitivity and wider linearity for thrombin detection between 0.0001 nM and 80 nM with a LOD of 50 fM, combining the aforementioned multiple effects [31]. 

Eguílaz et al. reported a bioanalytical platform based on a hybrid nanomaterial involving MWCNT and Hb-functionalized mesoporous MCM-41 silica for nitrite and trichloroacetic acid, followed by chronoamperometry. The Hb was immobilized in the pores of MCM-41 silica NPs and supported on MWCNT to generate the MWCNTs-MCM41-Hb hybrid bioconjugate, as shown in Figure 2I. The electrochemical biosensor exhibited a rapid response to the changes in NO_2_^−^ and trichloroacetic acid concentration with a wide linear range between 1.0 × 10^−7^ M and 1.25 × 10^−4^ M and a low LOD of 16 nM and a linear range of 5.0 × 10^−5^ M to 2.7 × 10^−2^ M with a LOD of 3 µM, respectively. The hybrid nanomaterial combined the advantages of each nanomaterial, the high surface area, biocompatibility, and protein loading capacity of MCM-41 NPs, and the high surface area and catalytic properties of MWCNTs promoting the direct electron transfer between Hb and the electrode surface [62]. Interestingly, Shekari et al. designed a unique and selective electrochemical aptasensor based on aptamer-glutaraldehyde-amino-functionalized MCM-41-glassy carbon electrode (Ap-GA-NH_2_MCM-41-GCE) for the detection of hemin and hemoglobin (Hb). The authors’ anchorage used glutaraldehyde (GA) as a linker agent and an aptamer with a sequence (5′-NH_2—_GTGGGTAGGGCGGGTTGG-3′) onto the NH_2_MCM-41-GCE. The aptasensor revealed a linear range of 1.0 × 10^−19^ to 1.0 × 10^−6^ M for both (hemin and Hb) with LOD of 7.5 × 10^−20^ M and 6.5 × 10^−20^ M, respectively [63]. 

Cai et al. reported the application of an ultrasensitive sandwich-type electrochemical biosensor to detect breast cancer (BRCA1). Horseradish peroxidase (HRP) was entrapped in the pores of amino-group-functionalized SBA-15 and the secondary antibody (Ab2) was linked to SBA-15 by covalent bonds. Ionic liquid (IL) was added to the mixed solution of SBA-15/HRP/Ab2; the IL increased the electrochemical activity of HRP and promoted electron transport (Figure 2II). The resultant immunoassay signal based on the amino-group-functionalized SBA-15 as label and HRP as enhancer improved the performance in the detection of BRCA1. Under optimal conditions, the electrochemical immunoassay exhibited a wide linear range from 0.01 to 15 ng/mL with 4.86 pg/mL BRCA1 LOD [64]. 

Recently, metallic nanomaterials have also been incorporated into siliceous structures. For example, Huang et al. introduced an electrochemical aptasensor based on gold NPs (AuNPs) doped with MCM-41 (Au-MCM-41) to detect codeine by differential pulse voltammetry (DPV). They utilized a DNA Apt (HL7-1 to HL7-18) anchored to a hybrid silicon nanostructure for specific monitoring of codeine, as shown in Figure 2III. The electrochemical nanobiosensor showed a linear range between 10 pM and 100 nM with a LOD of 3 pM [65]. Moreover, Roushani et al. developed an electrochemical nanobiosensor to determine streptomycin (STR) by voltammetry. An STR-specific aptamer was immobilized on a gold electrode modified with a mesoporous silica thin film (MSF) that was functionalized with (3-aminopropyl) triethoxysilane (APTES) and silver nanoparticles (AgNPs) (Figure 2IV). The aptasensor, which works best with a working potential of 0.22 V (versus Ag/AgCl), showed a linear response in the STR concentration range from 1 fg/mL to 6.2 ng/mL with a LOD of 0.33 fg/mL [66]. Another study conducted by Shekari et al. developed an impedimetric biosensor based on AuNPs loaded in functionalized mesoporous silica nanoparticles (MSNPs) for carcinoembryonic antigen (CEA) biosensing. The GCE was modified by AuNPs merged in amino-functionalized MCM-41; afterwards, aptamer was immobilized via a covalent bond. The impedimetric biosensor showed a linear range of 1.0 × 10^−3^ to 100 ng/mL with a lower LOD of 9.8 × 10^−4^ ng/mL [67].

### 2.3. Carbon Nanostructures

Carbon-based nanomaterials have attracted increasing attention in electrochemical nano(bio)sensor development owing to their unique properties [68]. Among the carbonaceous nanomaterials, graphene and carbon nanotubes (CNT) have received considerable interest because they can be used simultaneously as transducer platforms, as matrixes to immobilize bioreceptors, and as signal amplification systems [46,69]. Different allotropes of carbon exist, such as graphite, diamond, CNT, and fullerenes [70]. The confined sp^2^ hybridized orbitals arranged in a hexagonal lattice in graphene facilitate high electric conductivity and zero-gap linear electron momentum dispersion, essential in electrochemical devices. 

Graphene-based nanomaterials have emerged as a new field of research due to their unique properties, such as high electron transport capability, electric conductivity, mechanical strength, quantum Hall effect at room temperature, and tunable bandgap. Furthermore, graphene is a nanomaterial of low cost and substantial green environmental impact, making it suitable for biosensing applications [70,71,72]. Graphene-based electrochemical biosensors’ specificity lies in the fact that different molecules oxidize or reduce at a different potential window, where heterogeneous electron transfer for oxidoreduction processes occurs between the graphene electrode and the electroactive analyte molecules or probes. These processes occur at the edges and corners of the graphene or the defect sites of its basal plane. In this context, the high surface area of graphene provides an enormous number of corners, edges, and defects, which can act as superior electroactive sites that may improve the performance of the resultant platforms [72,73]. 

Graphene mainly has four forms, i.e., graphene itself; the oxidized form of chemically modified graphene (GO); the reduced form of graphene oxide (rGO); and graphene quantum dots (GQD), which have a size less than 10 nm [68,74,75,76]. There are several reports about the synthesis of graphene, including chemical vapor deposition (CVD) [77], plasma-enhanced chemical vapor deposition (PE-CVD), cleavage of natural graphite, arc discharge, epitaxial growth of electrically insulating materials, solution-processable methods, and the microwave-assisted synthesis method [47]. Whereas GO can be synthesized by rapid oxidation of crystalline graphite, followed by some dispersion methods in various organic solvents, the high temperature under reducing conditions converts GO to rGO [78,79]. GQDs are a 0D material with sizes below 10 nm and characteristics deriving from graphene and carbon dots. GQDs can be readily synthesized by pyrolyzing citric acid and dispersing the carbonized products into alkaline solutions [80], exfoliation [81], and hydrothermal methods [82], among others [80]. 

Although graphene exhibits unique 2D structural, chemical, and electronic properties, these only emerge in the 2D planar direction, limiting its scope and application. Moreover, some inherent disadvantages of pristine graphene, such as easy aggregation, and poor solubility and/or processability, represent significant obstacles in developing electrochemical biosensors [83]. Therefore, to optimize and further expand the use of this 2D nanomaterial, new efforts have attempted to address these weaknesses by developing structures wherein graphene acts as a scaffold for anchoring other nanomaterials. Apart from taking full advantage of the superior properties of graphene and the corresponding functionalizing nanomaterial, these graphene hybrids are also endowed with new desirable properties [84]. 

Among carbon nanotubes (CNT), single-wall carbon nanotubes (SWCNT) possess the cylindrical nanostructure of a single graphite sheet and MWCNT comprises multiple superimposed rolled graphite layers. The graphite layers are composed of conjugated sp^2^-hybridized carbon atoms arranged into a planner 2D honeycomb-like lattice [85]. CNTs can be synthesized by arc-discharge [86], laser ablation [87], and CVD [88,89]. Arc-discharge is straightforward if two graphite electrodes are introduced. However, many side products are formed simultaneously, requiring a postdeposition purification process that increases production costs. Direct laser vaporization involves a pulsed or continuous laser beam introduced into a 1200 °C furnace to vaporize a target made of graphite and metal catalysts (cobalt or nickel). From this method, CNT of relatively high purity can be synthesized by adjusting the nature of the gas and its pressure [42]. Synthesis by CVD is a better technique to produce CNT matrices of higher performance and purity at moderate temperature, which involves decomposing carbon gas sources. The thermal CVD process involves a catalyst preparation step followed by the actual synthesis of the nanotubes. A higher CVD temperature positively affects the crystallinity of the produced structures and may also result in higher growth rates. However, there are great concerns about the impurities in the CNT synthesized by the previous methods. Although acid treatments typically remove impurities, these treatments, in turn, can introduce other types of impurities, which may affect the physicochemical properties of nanotubes [90,91]. CNTs have been widely used in the development of electroactive platforms due to their unique properties such as electrical conductivity; relative biocompatibility; surface functionality; high surface area; high adsorption capacity; wide potential window; high mechanical, thermal, and chemical stability; and low background current [68,92]. In addition, CNT-based platforms have demonstrated higher sensitivity and better performance due to their 1D π-electron conjugation through hollow cylinders, which is responsible for the efficient capture and promotion of electron transfer from analytes and makes them very attractive in many applications. However, the electrical conductivity of CNTs is affected by the nonuniform contact, discontinuities, and defects of nanomaterials generated by the synthesis process, thereby limiting their applications in biosensors [88]. 

Given the high interest and broad scope of this type of hybrid nanomaterial, we discuss recent advances in electrochemical biosensing platforms based on hybrid-CNT for clinical biomarkers. For example, different authors have used carbon nanotubes to prepare hybrid materials. Hossain et al. developed a hybrid glucose biosensor based on glucose oxidase immobilized on PtNPs decorated chemically derived graphene (CG) and carbon nanotube electrode platform for glucose sensing by chronoamperometry. The synergy between PtNPs and CG/MWCNTs enhanced the electrocatalytic activity and allowed a selective glucose detection exhibiting a linear range of 0.5 mM to 13.5 mM with a LOD of 1.3 μM [93]. Keerthi et al. designed a molybdenum nanoparticle self-supported functionalized MWCNT (MoNPs@f-MWCNTs)-based core-shell hybrid nanomaterial for electrochemical detection of dopamine (DA) by electrochemical impedance spectroscopy (EIS), as shown in Figure 3I. The impedimetric biosensor showed a linear response from 0.01 µM to 1609 µM with a LOD of 1.26 nM. These results demonstrated that the Mo NPs@f-MWCNTs hybrid material possesses tremendous superiority in DA detection mainly due to the large surface area and numerous electroactive sites [94]. 

Aguílaz et al. reported a hybrid material consisting of the conducting polymer poly-(3-methylthiophene) (P3MT) and MWCNT for cytochrome c (Cyt c) detection by chronoamperometry. The combination of the conductivity and high surface area of the polymer and CNT in the hybrid material promoted the electron transfer in the electrochemical biosensor. Furthermore, it facilitated the formation of self-assembled monolayers of Cyt c mediated by l-cysteine (l-Cys) on P3MT/MWCNT/GCE at neutral pH, where positively charged Cyt c strongly interacted with negatively charged sites of l-Cys, as shown in Figure 3II. Finally, the electrochemical biosensor exhibited a linear range of 0.7 and 400 μM, with a LOD of 0.23 μM [95].

Other CNT-nanostructures employed in developing electrochemical biosensors are based on GQD. As an illustration, Serafin et al. designed the first integrated electrochemical immunosensor for the determination of IL-13Rα2. The strategy used a hybrid nanomaterial composed of MWCNTs and GQDs as nanocarriers of multiple detector antibodies and HRP molecules (Figure 3III). As a result, amperometric detection with the H_2_O_2_/hydroquinone (HQ) system achieved a linear range from 2.7 to 100 ng/mL IL-13sRα2, with a low LOD of 0.8 ng/mL [96]. Further, Hasanzadeh et al. reported an ultrasensitive electrochemical immunosensor for quantitation of the p53 tumor suppressor protein by immobilizing a hybrid nanostructure containing poly l-cysteine (P-Cys) as conductive matrix and GQDs/AuNPs (GNPs) as dual amplification elements. Under optimized conditions, the calibration curve for p53 concentration was linear from 0.000197 to 0.016 pM by square wave voltammetry (SWV) technique and 0.195 to 50 pM by DPV with a low limit of quantification (LOQ) of 0.065 fM. Besides, the linear range of p53 0.000592 to 1.296 pM and limit of quantification 0.065 fM were lower in unprocessed human plasma [97]. Buk et al. developed a carbon quantum dots (CQDs)/AuNPs nanohybrid material to be applied as an immobilization matrix of Gox enzyme. The cheapness and versatility of the CQDs were directly combined with the inertness and electrochemical activity of the AuNPs to create a new electrochemical biosensor for glucose detection by chronoamperometry. The CQDs/AuNPs-GOx biosensor exhibited a sensitivity of 47.24 μA/(mM·cm^2^) and a LOD of 17 μM with a linear response from 0.05 mM to 2.85 mM [98]. Besides, Liu et al. designed a novel heterogeneous architecture integrating two-dimensional (2D) bimetallic CoCu–zeolite imidazole framework (CoCu-ZIF) and zero-dimensional (0D) Ti_3_C_2_Tx MXene-derived carbon dots (CDs) (represented by CoCu-ZIF@CDs) for anchoring B16-F10 cell-targeted aptamer strands and detecting B16-F10 cells from the biological environment. Compared with CoCu-ZIF- and CD-based cytosensors, the constructed CoCu-ZIF@CD-based one showed superior sensing performance, with an extreme LOD of 33 cells/mL and a range of suspension concentrations from 1 × 10^2^ to 1 × 10^5^ cells/mL B16-F10 cells [99].

### 2.4. Polymers

Polymeric nanostructures provide outstanding support for the immobilization of bioreceptors in preparing electrochemical biosensors due to their excellent biocompatibility, high affinity, strong adsorption ability, low molecular permeability and physical rigidity, and chemical inertness in biological processes [100]. Polymeric nanostructures generated by the electropolymerization process have an organized molecular structure, with multiple functional groups and linking spots for immobilizing the bioreceptors, thus providing a suitable environment for their oriented anchorage and preserving their activity for a long-term duration [101]. Nanobioengineered polymer structures have recently aroused great interest as potential candidates for improving electrochemical biosensors’ response time, sensitivity, and versatility [39,100]. The polymeric nanostructures are commonly membranes or fibers based on polyacetylene (PA), polypyrrole (PPy), polythiophene (PT), polyaniline (PANI), among others. They are easy to prepare and enjoy attractive unique properties such as high stability at room temperature, high conductivity output, easy polymerization, and high compatibility with biological molecules [101]. 

Polymer nanostructures such as PPy, PANI, and PT can be obtained by electrochemical polymerization [102] or electrospinning [103,104]. The morphology of the polymer films or nanofibers can be tailored by controlling the charge transferred during the polymerization process and other parameters such as temperature, monomer concentration, polymerization potential, current, time, and nature of the supporting electrolyte. Moreover, polymeric films decrease the number of interfering compounds due to their size-exclusion and ion-exchange characteristics. The π-electron backbone, an extended conjugated system with single and double bonds alternating along the polymer chain, is responsible for the unusual electronic properties of polymer nanostructures such as electrical conductivity and low energy optical transitions, low ionization potential, and high electron affinity [100]. 

Güner et al. reported an electrochemical immunosensor for the *Escherichia coli* food pathogen by modifying a pencil graphite electrode surface with a hybrid nanobiocomposite. The authors prepared a pyrrole branched chitosan (Chi-Py) mixture, AuNPs, and MWCNT dropped on the electrode surface. Subsequently, the monoclonal anti-*E. coli* O157:H7 antibodies were linked to amino groups from Chi-Py with glutaraldehyde as the binding agent, as shown in Figure 4I. This hybrid nanobiocomposite-based immunosensor demonstrated high selectivity towards the Gram-negative pathogenic species *E. coli* O157:H7 with a linear range from 3 × 10^1^ to 3 × 10^7^ cfu/mL and a LOD of ~30 CFU/mL in phosphate-buffered saline (PBS) solution. Furthermore, the enhanced properties of the nanohybrid structure promoted the anchoring of many antibodies in an oriented manner, offering a reliable means of quantifying the bacteria under study [105]. On the other hand, Tang et al. prepared a nanohybrid with a high synergistic effect that consisted of polyaniline/active carbon and nanometer-sized TiO_2_ (n-TiO_2_) by oxidation and sol–gel methods, respectively, which were then used as a zymophore of glucose oxidase (GOx) to modify a GCE. The electrochemical biosensor showed a proper response to glucose with direct electron transfer. The linear range of the detected glucose concentration was from 0.02 mM to 6.0 mM, the sensitivity was 6.31 μA/(mM·cm^2^), and the LOD was 18 μM [106]. 

Other polymeric nanostructures employed in developing electrochemical biosensors are based on poly(3,4-ethylenedioxythiophene) (PEDOT). For example, Chen et al. modified the surface of a GCE by electrochemical deposition of a mixture containing PEDOT-citrate, followed by subsequent AuNPs electrodeposition by the CV method. Finally, a Y-shaped peptide with two arms was immobilized on the GCE, one arm with the EKEKEKE peptide sequence for antifouling and the other arm with the HWRGWVA peptide sequence for the recognition of human IgG in human serum. The GCE/PEDOT-citrate/AuNPs/Y-shaped peptide biosensor showed a linear range for human IgG detection from 0 pg/mL to 10 μg/mL, with a relatively low LOD of 32 pg/mL [109].

Further, Liu et al. developed a nanobiohybrid based on GOx-hybrid conducting polymer microspheres with PEDOT as polymer support for glucose detection by chronoamperometry. The biohybrid conducting polymer was prepared based on a template-assisted chemical polymerization leading to PEDOT microspheres (PEDOT-MSs), followed by in situ deposition of PtNPs and electrostatic immobilization of GOx to form GOx-PtNPs-PEDOT-MSs nanobiohybrid (Figure 4II). The biosensor showed a linear response towards glucose over 0.1–10 mM with a LOD of 1.55 µM. This new nanohybrid combines the advantages of microstructure, morphology, the high active surface area of PEDOT with intrinsic biocatalytic activity, and conductivity properties of metallic particles, thus demonstrating its potential as a hybrid nanomaterial suitable for the development of electrochemical biosensors [107].

Khalifa et al. established a PANI/AuNPs hybrid modified electrode for the rapid detection of pyocyanin (PYO) in *Pseudomonas aeruginosa* infections by SWV, as shown in Figure 4III. The hybrid combines the high electrical conductivity of the polymer with the high affinity of the AuNPs for the union of biomolecules. It generated a nanohybrid with the ability to act as a scaffold for the immobilization of biological species. The amperometry biosensor exhibited a linear range from 1.9 μM to 238 μM and a detection limit of 500 nM [108]. Similarly, Hui et al. loaded AuNPs upon PANI nanowires, which were then functionalized with polyethylene glycols (PEG-NH_2_) and specific antibodies for alpha-fetoprotein biosensing by DPV. Using the redox current of PANI as the sensing signal, in addition to the excellent biocompatibility of PEG/AuNPs and the anti-biofouling property of PEG, the developed immunosensor showed a wide linear range between 10^−14^ to 10^−6^ mg/mL and ultralow LOD of 0.007 pg/mL [110]. 

### 2.5. Emerging Nanostructured Nanomaterial for Sensitive Biosensing

The hybrid nanomaterials based on transition metal dichalcogenides (TMD) such as MoS_2_, WS_2_, MoSe_2_, WSe_2_, and MoTe_2_ have received much attention, mainly due to their peculiar, layered structures and consequent ability to form 2D semiconductors. In addition, TMD monolayers display particular optoelectronic properties, such as a direct bandgap and second-harmonic generation. Recently, Dou et al. established a nanoflower-decorated trimetallic hybrid MoS_2_ nanosheet-modified sensor to monitor H_2_O_2_ secreted by living MCF-7 cancer cells. The MoS_2_ nanohybrids dispersed in Au-Pd-Pt nanoflowers were synthesized by a simple wet chemistry method. The resulting nanohybrids exhibited significantly enhanced catalytic activity towards the electrochemical reduction of H_2_O_2_ due to the synergistic effect of the highly dispersed trimetallic hybrid nanoflowers and MoS_2_ nanosheets, resulting in the ultrasensitive detection of H_2_O_2_ with a LOD of sub-nanomolar level in vitro [111]. 

Wang et al. reported an ultrasensitive biosensor to determine thrombin using an electrode modified with WSe_2_ and AuNPs, aptamer-thrombin-aptamer sandwiching, redox cycling, and signal enhancement alkaline phosphatase. The AuNPs were linked to thrombin aptamer 1 via Au-S bonds. Thrombin was first captured by aptamer 1 and then sandwiched through the simultaneous interaction with AuNPs modified with thrombin-specific aptamer 2 and signaling probe. Subsequently, the DNA-linked AuNP hybrids captured streptavidin-conjugated alkaline phosphatase onto the modified GCE through the specific affinity reaction for further signal enhancement. As a result, the electrochemical biosensor showed a linear range from 0–1 ng/mL and the LOD as low as 190 fg/mL [112]. 

Other nanohybrids used in preparing electrochemical biosensors with improved performance are those based on transition metal carbides and nitrides (MXenes). For example, Liu et al. modified the surface of a GCE with layers of MoS_2_/Ti_3_C_2_ nanohybrids prepared by a hydrothermal method for micro-ribonucleic-acid (miRNA) detection by DPV. In particular, this biosensing platform showed a high linear detection window ranging from 1 fM to 0.1 nM with a LOD of 0.43 fM [113]. Yang et al. developed an electrochemical biosensor based on AuNPs/T_i3_C_2_ MXene 3D nanocomposite for microRNA-155 detection by exonuclease III-aided cascade target recycling. The 3D structure of the AuNPs/Ti_3_C_2_ MXene nanohybrid for the biosensor platform leverages the integrated advantages of a large specific surface area, excellent electrical conductivity, and electrocatalytic properties. The electrochemical biosensor achieved structural porosity and polar functional groups, the response of which was linear in a range from 1.0 fM to 10 nM with a LOD of 0.35 fM [114]. Chansi et al. assembled an immunosensor by concatenating nanoimmunohybrids and polyclonal antibodies (rIgG) (Chi-AuNP-rIgG-BSA) with a MOF for the detection of a wide range of pesticides (pyrethroids, neonicotinoids, and organophosphates). The developed BSA/Chi-AuNP-rIgG-BSA/MOF/ITO platform could measure total pesticide loading using CV at 0.41 V in a solution of [Fe(CN)_6_]^3−/4−^ as a mediator in PBS within a linear range from 4 to 100 ng/L, a sensitivity of 25.4 μA/(ng·cm^2^), and a LOD of 4 ng/mL [115]. Comparative analytical characteristics of nanomaterial-based electrochemical biosensors focused on the last ten years are summarized in Table 1.

**Table 1 molecules-27-03841-t001:** Comparative analytical characteristics of nanomaterial-based electrochemical biosensors focused on the last ten years.

Nanomaterial	Hybrid ^a^	Target ^b^	Analytical Characteristics		Comments	References
			Linear Range	LOD		
Metallic nanostructures	3D hybrid graphene–GNR.	H_2_O_2_	0 to 50 mM	2.9 µM	Metallic nanostructures have high catalytic activity, easy preparation, and relatively low cost. However, this kind of nanomaterial can change its oxidation state due to variations in conditions of the medium, such as pH, ionic strength, and temperature upon time.	[48]
	TiO_2_ nanoparticles encapsulated ZIF-8	Glucose	2 to 10 mM	80 nM		[51]
	Nanohybrid of VS_2_/AuNP and CoFe_2_O_4_ nanozyme	Kana	1 pM to 1 μM	0.5 pM		[52]
	Ag and hybrid Ag–Fe_3_O_4_ metallic nanoparticles.	AA	0.2–60 μM	74 nM		[49]
Silicon nanomaterials	mSiO_2_@MWCNT.	Thrombin	0.0001 nM and 80 nM	50 fM	These nanomaterials have high mechanical resistance, thermal stability, long functional life, and versatility; nonetheless, they require long synthetic processes, and their application is limited to certain analytes.	[31]
	MSF/APTES/AgNP	STR	1 to 6.2 ng/mL	0.33 fg/mL		[66]
	Ap–GA–NH_2_MCM-41–GCE	hemin and Hb	1.0 × 10^−19^ to 1.0 × 10^−6^ M	7.5 × 10^−20^ M and 6.5 × 10^−20^ M		[63]
	AuNPs loaded in functionalized MSNPs	CEA	1.0 × 10^−3^ to 100 ng/mL	9.8 × 10^−4^ ng/mL		[67]
Carbon nanostructures	MWCNTs and GQDs.	IL-13Rα2	2.7 to 100 ng/mL	0.8 ng/mL	These nanomaterials enjoy thermal stability, large surface area, and a wide range of nanostructures and functional groups. They are the main nanomaterials used in the preparation of electrochemical biosensors.	[96]
	GQDs/AuNPs.	P53	0.000592–1.296 pM	0.065 fM		[97]
	CQDs/AuNps	Glucose	0.05 mM to 2.85 mM	17 μM		[98]
	CoCu-ZIF@CDs	B16-F10 cells	1 × 10^2^ to 1 × 10^5^ cells/mL	33 cells/mL		[99]
Polymers	(Chi-Py) mixture, AuNPs, and MWCNT	Escherichia coli	3 × 10^1^ to 3 × 10^7^ cfu/mL	~30 CFU/mL	These have high biocompatibility, high affinity, strong adsorption ability, low molecular permeability, physical rigidity, and chemical inertness in biological processes. However, functionalizing their surface is necessary for the anchorage of bioreceptors, and some polymers oxidize due to changes in medium conditions.	[105]
	PANI/active carbon and n-TiO_2_	Glucose	0.02 mM to 6.0 mM	18 μM		[106]
	PEG/AuNPs/PANI	alpha-fetoprotein	10^−14^ to 10^−6^ mg/mL	0.007 pg/mL		[110]
Other nanostructured nanomaterials	WSe_2_ and AuNPs	Thrombin	0–1 ng/mL	190 fg/mL	Other hybrid nanostructures have a large specific surface area, excellent electrical conductivity, and electrocatalytic properties.	[112]
	MoS_2_/Ti_3_C_2_ nanohybrids	miRNA	1 fM to 0.1 nM	0.43 fM		[113]
	AuNPs/Ti_3_C_2_ MXene 3D	miRNA155	1.0 fM to 10 nM	0.35 fM		[114]

^a^ GNR, graphene–gold nanorod; AuNPs, gold nanoparticles; Ap, aptamer; GA, glutaraldehyde; GCE, glassy carbon electrode; MSNPs, mesoporous silica nanoparticles; MWCNTs, multiwalled carbon nanotube; MSF, mesoporous silica thin film; APTES, (3-aminopropyl) triethoxysilane; AgNP, silver nanoparticles; CDs, carbon-dots; Chi-Py, pyrrole branched chitosan; PEG, polyethylene glycols; PANI, polyaniline. ^b^ AA, ascorbic acid; STR, streptomycin; miRNA; micro-RNA.

## 3. Conjugation of Nanohybrid Materials with Biomolecules

Biosensors can be label-free and label-based. Briefly, in a label-free mode, the detected signal is generated directly by the interaction of the analyzed (bio)material with the transducer. In contrast, label-based sensing involves chemical or biological compounds that act as labels, generating a detectable signal by analytical techniques such as colorimetry, fluorescence, and electrochemistry [116]. This section will focus on the diversity of bioreceptors anchored to nanohybrid materials and the surface chemistry to generate electrochemical biosensors with improved yields.

### 3.1. Bioreceptors

Biological receptors are biomolecules that bind to a specific ligand with a defined structure, commonly through bioaffinity interactions. These biological components are used when assembling nanobiosensors due to their high specificity, differentiating the target molecule from analogous counterparts and even isomers of the same molecule (Figure 5). Different bioreceptors can be anchored at electrochemical transducers to confer specificity to nanobioengineered devices. They can generally be classified into five major categories, i.e., enzymes, antibody/antigens, nucleic acids, cellular structures/cells, and biomimetic entities, as depicted in Figure 5.

#### 3.1.1. Protein Bioreceptors and Peptides

Proteins are responsible for many biological processes in cells and tissues, but any failure in their normal function may result in a type of pathology such as cancer considering them as appropriate biomarkers. For sensing proteins involved in cancer, Zhou et al. proposed an electrochemical immunosensor based on a primary antibody (Ab1) adsorbed on top of AuNPs electroplated at the surface of a GCE and a hydroxyl pillar[5]arene (HP5)@AuNPs@g-C_3_N_4_ (HP5@AuNPs@g-C_3_N_4_) hybrid nanomaterial bioconjugated with a secondary prostate-specific antigen (PSA) antibody (Ab2). The immunosensor was tested with different PSA target concentrations by DPV, having a linear response from 0.0005–10.00 ng/mL and a LOD of 0.12 pg/mL [117]. 

Enzymes are one of the biorecognition elements widely used in biosensor applications. They are mainly used to function more as labels than the actual bioreceptor. Enzymes are proteins formed by a 3D structure composed of peptides, with an active zone that confers specificity to the enzyme, capable of selectively catalyzing a (bio)chemical reaction [118]. Enzymes are the most used element of recognition in developing electroactive platforms due to their low cost, high availability in the market, and easy manipulation [119,120]. However, enzymes are prone to undergo denaturation or loss of their catalytic activity during immobilization processes or changes in environmental conditions [121]. For example, Tang et al. developed a nanohybrid consisting of polyaniline/active carbon and nanometer-sized TiO_2_ (n-TiO_2_) that was used as a zymophore of GOx to modify a GCE and detect glucose. The linear range of the detected glucose concentration was from 0.02 mM to 6.0 mM, the sensitivity was 6.31 μA/(mM·cm^2^), and the LOD was 18 μM [106]. 

Peptides with a linear or cyclic sequence of amino acids have been used extensively in nanobiosensing devices. Their highly sensitive, selective, and biocompatible secondary structure can be modulated by introducing modifications in the amino acids, thus favoring interactions with the target. For example, Hwang et al. reported a sensitive biosensor for detecting human norovirus by employing a recognition affinity peptide-based electrochemical biosensor. The authors chemically synthesized a series of cysteine-incorporated and substituted recognition peptides with isolated amino acids and immobilized them on a gold sensing layer to develop the biosensing interface, further interrogated by EIS. The LOD with Noro-1 as a molecular binder was found to be 99.8 nM for recombinant noroviral capsid proteins (rP2) and 7.8 copies/mL for human norovirus, demonstrating a high target sensitivity [122]. 

Antibodies, along with nucleic acids, are highly selective because the 3D structure of antibodies can be linked to a specific substance (antigen, small molecules) and nucleic acids to DNA (RNA) by complementary base pairing, respectively. Antibodies may be polyclonal, monoclonal, or recombinant depending on their properties and how they are synthesized. Each antibody consists of four polypeptides—two heavy and two light chains joined to form a “Y” shaped molecule. The amino acid sequence in the “Y,” composed of 110–130 amino acids, varies significantly among different antibodies and gives the antibody specificity for a binding antigen. The variable region includes the light and heavy chains [123]. 

In recent years, the number of publications focused on the development of immunosensors has increased. Although antibodies do not have catalytic properties, limiting their direct use in the construction of nanobiosensors [124], those labeled with enzymes or catalytic metal nanoparticles (nanozymes) have promoted their application [125,126,127,128]. For example, Yola et al. prepared a sandwich-type sensitive voltammetric immunosensor for breast cancer biomarker human epidermal growth factor receptor 2 (HER2) detection. The electrochemical immunosensor was developed based on gold nanoparticles decorated copper–organic framework (AuNPs/Cu-MOF) and quaternary chalcogenide with platinum-doped graphitic carbon nitride (g-C_3_N_4_); then, HER2 antibody and antigen HER2 protein were conjugated to AuNPs/Cu-MOF as biosensor platform. The developed immunosensor showed high sensitivity with a LOD of 3.00 fg/mL and a linear range 1.0–100.0 ng/mL [129]. 

Label-free immunosensors have emerged as an alternative to their labeled counterparts, often with comparable analytical performance. For instance, Mollarasouli et al. reported a new, label-free electrochemical immunosensor to detect tyrosine kinase (AXL) in human serum. The affinity reactions were monitored by measuring the decrease in the DPV response of the redox probe Fe(CN)_6_^3−/4−^. As a result, the prepared immunosensor exhibited an improved analytical performance concerning other electrochemical reported immunosensors with a broader range of linearity of 1.7 and 1000 pg/mL and a LOD of 0.5 pg/mL [130].

#### 3.1.2. Nucleic Acids, Aptamers, Cells, and Other Bioreceptors

The high specificity of the base pairs—adenine/thymine (or uracil) and cytosine/guanine-in DNA or RNA strands makes these macromolecules bioreceptors considerably remarkable in assembling ultrasensitive nanobiosensors. The electrochemical signal is recorded after the DNA(RNA) strands hybridize their complementary strands by Watson and Crick complementarity base pairing. However, it is commonly necessary to use marks to evidence the biorecognition event [100,131], similarly to antibodies. As a way of illustration, Alzate et al. reported a genosensor for the differential diagnosis of the zika virus in a sandwich-type format, the genosensor employed antibody-digoxigenin-HRP (anti-Dig-HRP) as a signal amplifier and chronoamperometry for the measurements, showing a LOD of 0.7 pM and linear range of 5 to 300 pmol/L [132]. In the context of label-free nanogenosensing devices, a nanobioengineered tool based on a sandwich-type genosensor consisting of AuNPs linked to single-strand DNA (ssDNA) for the electrochemical detection of a Zika virus was recently developed by our group. The genosensor was assembled at a screen-printed gold electrode decorated with hierarchical gold nanostructures, with Ru^3+^ as an electrochemical reporter. The genosensor response, interrogated by DPV, showed a linear range from 10 to 600 fM with a LOD of 0.2 fM [133]. Other approaches explore nanoprobes based on mixtures of metallic nanomaterials with H_2_O_2_ mimicking enzyme properties. For example, Ye et al. synthesized a gold–silver core-shell (Au@Ag)-loaded iron oxide (Fe_3_O_4_) nanocomposite (Fe_3_O_4_-Au@Ag) as a label of signal DNA probe (sDNA) for the detection of cauliflower mosaic virus 35S (CaMV35S) gene. Under the optimal experimental conditions, the nanobiosensor responded linearly in the range of 1 × 10^−16^ to 1 × 10^−10^ M with a LOD of 1.26 × 10^−17^ M [134]. 

In cell-based bioreceptors, biorecognition is either based on a whole-cell/microorganism or a specific cellular component capable of binding to certain species. However, cells are sensitive to a wide range of biochemical stimuli, which hinders the selective detection of a biochemical recognition event. To illustrate, Zhou et al. reported a new impedimetric biosensor based on CNT and T2 bacteriophages for the rapid and selective detection of *Escherichia coli B*. The T2 bacteriophage (virus) was immobilized on polyethyleneimine (PEI)-functionalized carbon nanotube transducer on GCE. The changes in the interfacial impedance due to the binding of *Escherichia coli B* to T2 phage on the CNT-modified electrode surface was monitored by EIS. The highly selective detection showed a linear range between 10^3^ to 10^7^ CFU/mL with LOD of 1.5 × 10^3^ CFU/mL [135]. 

Glycans are emerging bioreceptors consisting of monosaccharides linked by glycosidic bonds. In glycoproteins and glycolipids, the reducing end of a glycan is covalently linked to amino acids or lipids, respectively. Glycans in such glycomolecules can be recognized by glycan-binding proteins or lectins to promote the adhesion among cells and the extracellular matrix and promote related cell signaling and migration. Glycoprotein glycans and membrane glycolipids can indirectly affect cell adhesion and signaling [136]. Some advantages that glycans present in nanobiosensing are associated with their physicochemical characteristics, i.e., higher stability compared to proteins and antibodies. Glycans are more susceptible to modifications during biosynthesis because they are metabolic products; therefore, minor alterations in the biosynthetic process cause high variations in their structure, which favors their detection [137,138,139]. To illustrate the glycan-based nanobiosensing concept, Echeverry et al. employed a glycosylphosphatidylinositol as a bioreceptor immobilized on screen-printed gold electrodes through a linear alkane thiol phosphodiester for differential detection of the protozoan parasite *Toxoplasma gondii*. The resultant device showed a linear trend in serum ranging from 1.0 to 10.0 IU/mL, with a LOD of 0.31 IU/mL, which is of clinical relevance, and results agreed with those from the gold standard based on fluorescence, with high reproducibility [140]. 

Synthetic biomimetic receptors are designed and fabricated to mimic an actual bioreceptor (antibody, enzyme, cell, or nucleic acids) to simulate the binding target molecules with similar affinities and specificities to natural receptors but much longer-term robustness and stability. For example, Dong et al., used metallic nanoparticles (MNPs) functionalized with aptamer-DNA to detect tumor exosomes extracted from lymph node carcinoma of a prostate cell line. First, the aptamer-DNA hybridized with three different multimessenger DNA strands forming nanobioconjugates with long DNA sequences. Then, the liposome was bound with MNP-aptamer-DNA, releasing the multimessenger DNA strands immobilized on top of a gold electrode (GE) under a hybridization step with a capture probe. Finally, the formed double-stranded DNA was cleaved by a restriction enzyme, producing an electrochemical response registered by DPV. The as-developed exosome biosensor exhibited a LOD of 70 particles/μL [141] and could be implemented for detecting exosomes in a complex biological system. In the same context, You et al. developed an ultrasensitive biosensor based on molecularly imprinted polymers (MIPs) to detect a glycoprotein assisted by SiO_2_ NPs decorated with AuNPs, functionalized with mercaptophenylboronic acid (MPBA) as bioreceptor and ferrocenylhexanethiol (Fc) as an electrochemical reporter. A GCE was modified with chitosan and AuNPs-rGO, followed by immobilization of the target glycoprotein and the specific interaction with Fc/MPBA/AuNPs-SiO_2_ nanobioconjugate. At optimal conditions, the biosensor response tested by DPV was linear from 1 pg/mL to 100 ng/mL and reached a LOD of 0.57 pg/mL [142].

## 4. Functional Groups and Conjugation Chemistry

Assembling nano(bio)sensors involves binding the bioreceptor’s specific and oriented form to the transducer surface [143] by physical or chemical methods. The physical methods include the following: (i) physical adsorption of the bioreceptor on a matrix based on hydrophobic, electrostatic, and van der Waals attractive forces; (ii) enzyme entrapment in a sol–gel, hydrogel, or paste, confined by semipermeable membranes; (iii) encapsulation—confinement of the biomolecule within a solid matrix. The chemical immobilization methods include (i) covalent binding of the bioreceptor to a solid matrix or directly to the surface of the transducer; (ii) crosslinking employing multifunctional, low-molecular-weight reagents based on the formation of strong covalent bonding between the transducer and the biological material using a bifunctional agent; (iii) affinity binding, exploiting specificity of a bioreceptor to its support under different physiological conditions (Figure 6). Conjugation is achieved either by coupling the bioreceptor to the matrix based on affinity interactions or conjugating the bioreceptor to an entity that develops affinity toward the matrix [144]. 

The physical methods used in developing nano(bio)sensors are based on weak interactions and, therefore, the most straightforward and affordable. However, they are affected by environmental conditions such as pH, temperature, and ionic strength, generating biomolecule leaching processes during biodetection and storage. For this reason, covalent binding is commonly used as an immobilization alternative. However, this methodology requires modifying the surface of the electrode with a specific functional group that includes carboxylic acid (-COOH), aldehyde (-CHO), amine (-NH_2_), sulfhydryl (-SH), and azide (-N_3_) for the anchorage of bioreceptors. Therefore, this methodology requires precise knowledge of the surface chemistry of the electrode surface and the bioreceptor to favor the specific and oriented anchoring of the biomolecules [42]. 

Modification of the surface of the transducer can be achieved by bifunctional agents such as 4-aminobenzylamine (ABA) [145], 1-pyrenebutanoic acid, acid/basic treatment, or by modification of the electrode surface with affinity agents such as cysteamine [132], cyclodextrin [146], and biotin-avidin [38]. Finally, the modulation of bioreceptor–platform interactions [25] through specific groups on the nanostructured surface and the bioreceptors determines the methodology of immobilization, promotes the bioreceptor orientation, and increases its compatibility with the platform interface, thus influencing the selectivity, specificity, and stability of the resultant nanobioengineered platforms [28,143,146,147]. Therefore, changes in the surface chemistry of the platforms influence the physicochemical and electrochemical properties of the resultant (bio)sensing devices [148,149]. 

Bioreceptors can be reversibly adsorbed or trapped and retained or embedded on the surface of electrochemical platforms through ionic, electrostatic, hydrogen bonding, hydrophobic, or van der Waals interactions or irreversibly through covalent bonds. The interactions depend not only on the morphology and reactive functional groups on the electrochemical platform but also on the chemical nature, affinity, isoelectric point, and polarity of the solvent as well as the medium conditions for bioreceptor anchoring. As an illustration, the physical adsorption of bioreceptors can show low reproducibility due to the leaching effect during the analysis and little stability in different medium conditions. In contrast, the binding of biomolecules by covalent bonds through activated functional groups, often including carboxylic acid, amino, thiols, and esters, offers high stability despite possible aggregation, polymerization, and random biomolecule orientation [150]. Conjugation of nano(bio)sensors involves binding the bioreceptor’s specific and oriented form to the transducer surface by physical or chemical methods, as depicted in Figure 6. 

## 5. Characterization of Nanobioengineered Platforms

Some of the main techniques to characterize nanoengineered platforms include electrochemical and physiochemical techniques such as CV, EIS, and DPV to characterize electrochemical behavior and electron transfer; Fourier transform infrared spectroscopy (FTIR) to characterize the composition and surface chemistry; scanning electron microscopy (SEM) to characterize morphology and composition; and dynamic light scattering (DLS) to determine the surface properties, summarized in Table 2 [151].

## 6. Examples of Nanobioengineered Platforms for Electrochemical Biosensing in the Last Five Years

It has been reported that the construction of modified electrodes with hybrid or nanocomposite materials based on carbon nanomaterials doped with metallic nanoparticles [27,28,152,153,154] and semiconductor polymers [35,36,37,38,39] has great potential for the development of biosensors with enhanced performance (Table 3). For example, an immunosensor based on cadmium selenide (CdSe)-QD-melamine has been explored to detect CEA. The antigen was sandwiched between an Ab1-TiO_2_-AuNP-ITO (ITO-Indium tin oxide) transducer platform and an Ab2-QD-based network, producing linear photocurrent signal responses after photostimulation, with a linear range from 0.005 to 1000 ng/mL with LOD of 5 pg/mL [155]. Zhang et al. developed a highly sensitive electrochemical nanobiosensor to detect a specific sequence of miRNAs based on a triple signal amplification strategy. The electrochemical biosensor combined duplex-specific nuclease (DSN)-assisted target recycling joint with the conductive capacity of AuNPs and catalytic capacity of HRP for signal amplification. The electrochemical biosensor interrogated by chronoamperometry detected miRNA-21 with 43.3 aM LOD. The biosensor could discriminate miRNAs sequences with one base mismatch, showing this methodology’s great potential for biomedical applications [156]. Further, Liu et al., developed a hydrogel electrochemical biosensor based on ITO/polyethylene terephthalate (ITO/PET) to detect cancer-specific miR-21. Ferrocene-tagged recognition probes were cross-linked with DNAs grafted on the polyacrylamide backbones to form the hybrid DNA hydrogel, which was further immobilized on 3-(trimethoxysilyl)propyl methacrylate-treated ITO electrode. When the recognition probe was hybridized with the target miR-21, the hydrogel dissolved, producing a loss of ferrocene tags and a reduction in current, detected by DPV. The electrochemical biosensor is capable of detecting miR-21 at a concentration as low as 5 nM and linear read-out from 10 nM to 50 μM [157]. 

Hybrid nanomaterials can be obtained in many ways and present various morphologies. For example, Baek et al. fabricated a copper (Cu)-nanoflower decorated AuNPs-GO nanofiber for glucose detection by CV. The GO was mixed with poly(vinyl alcohol) (PVA) to generate nanofibers through electrospinning, then incorporated organic–inorganic hybrid nanoflower (Cu nanoflower-GOx and HRP). The electrochemical characterization revealed that Cu-nanoflower@AuNPs-GO NFs exhibited outstanding electrochemical catalytic nature and selectivity for converting glucose to gluconic acid in the presence of GOx-HRP-Cu nanoflower. The Cu-nanoflower@AuNPs-GO NFs-modified Au chip exhibited higher catalytic properties, which are attributed to the coating of unique organic–inorganic nanostructured materials on the surfaces of Au chip. The biosensor exhibited a linear range between 0.001 to 0.1 mM, with a LOD of 0.018 μM [158]. Jothi et al. designed a nonenzymatic biosensor based on graphene sheet/graphene nanoribbon/nickel nanoparticles (GS/GNR/Ni) for glucose detection by chronoamperometry. The different graphene structures generated a greater surface area for the incorporation of NiNPs; in turn, the hybrid showed a high electrochemical activity towards glucose oxidation in a 0.1 M NaOH solution. The obtained biosensor showed a linear range of 5 nM to 5 mM with a LOD of 2.5 nM [159]. Wang et al. presented a label-free electrochemical immunosensor for sensitively detecting carbohydrate antigen 19-9 (CA19-9) as a cancer biomarker. The authors employed a series of bimetallic cerium and ferric oxide nanoparticles embedded within the mesoporous carbon matrix (represented by CeO_2_/FeOx@mC). The CA 19-9 antibody can be anchored to the CeO_2_/FeOx@mC network through chemical absorption by esterlike bridging between the carboxylic groups of antibodies and CeO_2_ or FeOx. The EIS results showed that the developed immunosensor exhibited a LOD of 10 μU/mL linearly from 0.1 mU/mL to 10 U/mL toward CA 19-9 [160]. 

Combining different MOFs into a conjugate material can integrate the properties of each MOF component and further lead to emergent properties from the synergistic heterostructured units. As an illustration, Wang et al. designed an electrochemical platform based on Tb-MOF-on-Fe-MOF conjugate for ultrasensitive detection of carbohydrate antigen 125 (CA125) living in Michigan cancer foundation-7 (MCF-7) cells by EIS. Although the integrated MOF-on-MOF architectures show similar chemical and structural features to the top layer, the Fe-MOF-on-Tb-MOF and Tb-MOF-on-Fe-MOF have different surface nanostructures to their parent MOFs. The developed aptasensor based on Tb-MOF-on-Fe-MOF displayed higher stability of the formed aptamer-CA125 G-quadruplex than that based on Fe-MOF-on-Tb-MOF, owing to the stronger affinity of the aptamer for the Tb-MOF-on-Fe-MOF nanomaterial. The developed aptasensor provides a LOD of 58 μU/mL towards CA125 within a linear range from 100 μU/mL to 200 U/mL. Moreover, the Tb-MOF-on-Fe-MOF nanoarchitecture demonstrated superior biocompatibility and high endocytosis. As a result, the developed aptasensor showed a high sensitivity for detecting MCF-7 cells with a LOD of 19 cells/mL [161]. Lu et al. employed gold nanoflowers (AuNFs) on molybdenum disulfide (MoS_2_)-nanosheet-supported magnetic MOF (MMOF) Fe_3_O_4_@ZIF-8 hybrid nanozymes for drug evaluation with in situ monitoring of H_2_O_2_ released from H9C2 cardiac cells by chronoamperometry. Fe_3_O_4_@ZIF-8 nanomaterial was selected because of the high peroxidase-mimicking activity of magnetic Fe_3_O_4_ nanoparticle and the large pore size and surface area of metal-organic framework ZIF-8. MoS_2_ nanosheets and one-step electrodeposition of gold nanostructures enhanced the long-term stability, conductivity, and catalytic performance. The obtained hybrid nanomaterial (AuNFs/Fe_3_O_4_@ZIF-8-MoS_2_) exhibited prominent electrocatalytic activity to reduce H_2_O_2_ with a linear detection range of 5 μM-120 mM and a LOD of 0.9 μM [162]. 

Li et al. presented a label-free electrochemical immunosensor based on chrysanthemum-like using Co-based MOF (Co-MOFs) as carriers and copper–gold nanowires (CuAu NWs) wrapped around their surface for the detection of nuclear matrix protein-22 (NMP-22). The Co-MOFs/CuAu NWs possessed outstanding catalytic capabilities, which served as signal materials and simultaneously carried the anti-NMP-22 antibody (Ab). The NMP-22 biosensor exhibited a linear range from 0.1 pg/mL to 1 ng/mL, with a LOD of 33 fg/mL [163]. Later, Ding et al. fabricated a novel electrochemical detection method for glucose using CuOx@Co_3_O_4_ core-shell nanowires on Cu foam substrate as a potential electrode. The as-fabricated hierarchical composite-MOF electrode exhibited the structural characteristics of CuOx nanowires as core and Co_3_O_4_ nanoparticles as a shell that calcinated from microporous ZIF-67. The constructed glucose sensor exhibited a linear range from 0.1 to 1300.0μM with a LOD of 36 nM [164]. Wang et al. coupled graphene layers, chitosan-transition metal carbides, and acetylcholinesterase on a GCE surface for organophosphate pesticides biosensing. The acetylcholinesterase/chitosan-transition metals/graphene/GCE nanobioconjugate was characterized by CV, DPV, and EIS and compared with biosensors without modification. The resultant platform showed a better catalytic performance due to the excellent electrical properties, biocompatibility, and high surface area of graphene and transition metal carbides nanobioconjugate. In addition, the electrochemical biosensor exhibited a linear response from 11.31 μM to 22.6 nM with LOD of 14.45 nM [165]. Kasturi et al. published a novel, high-resolution electrochemical biosensor using AuNP-dotted reduced graphene oxide (rGO/Au) for the detection of the miRNA-122 by DPV. The rGO/Au nanomaterial was prepared using green synthesis; then, the probe DNA was anchored onto the binding sites of rGO/Au through thiol linker and recognized the target miRNA-122, as shown in Figure 7I. The developed rGO/Au based electrochemical biosensor demonstrated a linear response for various target miRNA-122 concentrations with a range from 10 μM to 10 pM and a LOD of 1.73 pM [166]. Zou et al. employed ultrathin (UT)-g-C_3_N_4_/Ag hybrids for L-tyrosinase by DPV. The UT-g-C_3_N_4_/Ag hybrids were fabricated via a photoassisted reduction method. Electrochemical sensor showed a linear range of 1.00 × 10^−6^ to 1.50 × 10^−4^ mol/L with a LOD of 1.40 × 10^−7^ mol/L [167].

Another hybrid nanomaterial that has been used for the development of electrochemical biosensors is based on MXenes, as commented. For example, Koyappayil et al. reported that Ti_3_C_2_Tx nanosheets decorated with β-hydroxybutyrate dehydrogenase as a transducer platform for developing an electrochemical β-hydroxybutyrate biosensor with amperometric β-hydroxybutyric acid detection. The biosensor displayed a sensitivity of 0.480 μA/(Mm·cm^2^), a wide linear range from 0.36 to 17.9 mM, and a LOD of 45 μM [168]. Another breakthrough in developing peptide-based electrochemical biosensors was reported by Xu et al., who designed a universal strategy for constructing ratiometric antifouling electrochemical biosensors based on multifunctional peptides 2D MXene nanomaterials loaded with AuNPs and methylene blue (MB) to PSA detection. MXene provided a high conductivity to the biosensor; the AuNPs acted as anchoring support for the biomolecules containing terminal sulfhydryl and MB as the electrochemical probe. MXene-Au-MB nanohybrid was fixed on the Nafion electrode surface, and covalent bonds linked a multifunctional peptide-functionalized with thiol groups at its terminal end (Figure 7II). The biodetection system was dependent on PSA concentration in a detection range from 5 pg/mL to 10 ng/mL with a LOD of 0.83 pg/mL [169]. Li et al. presented a simple, robust, label-free homogeneous electrochemical detection platform for protein kinase activity detection and inhibition by integrating carboxypeptidase Y-assisted peptide cleavage reaction and MSF arranged vertically. The designed peptide was composed of a recognized kinase-specific sequence and multiple positively charged arginine residues to effectively adsorb to the negatively charged surface of the MSF-modified ITO electrode (MSF/ITO) by noncovalent electrostatic attraction. The authors used both protein kinase A (PKA) and casein kinase II (CK2) as biodetection models followed by chronoamperometry, where the LODs were 0.083 and 0.095 U/mL, respectively [170]. 

Point-of-care tests (POCT) based on nanobiohybrid materials have recently gained much attention due to their convenient, simple interrogation, and accessible features. To show such remarkable properties, Jędrzak et al. described a novel nanoplatform based on biomimetic polymer polynorepinephrine (PNE) grafted on magnetite nanoparticles (Fe_3_O_4_) with GOx from *Aspergillus niger* (Fe_3_O_4_@PNE-GOx). The system was integrated with a smartphone analyzer as a POCT biosensor for glucose measurement by chronoamperometry (Figure 7III). This portable electrochemical sensor could detect glucose between 0.24 to 24 mM with a LOD of 6.1 µM [171]. 

Electrochemical nanobiosensors are particularly suitable for miniaturization and integration in microfluidic devices. In this context, a suitable integration into a microfluidic system is required to ensure the device’s integrity, accessibility for the sample solution, and appropriate sample handling to exploit a biosensor’s full potential. Miniaturized electrochemical detection approaches have recently been described and compared in detail [172]. For example, Boopphahom et al. utilized a digital dispenser to deliver a copper oxide and ionic liquid composite onto an electrochemically reduced graphene-modified screen-printed carbon electrode (CuO/IL/ERGO/SPCE) on a paper-based analytical device (PAD) for creatinine monitoring by chronoamperometry. The authors employed a combination of advantageous nanomaterials and added a controllable deposition medium to produce a selective, rapid, disposable, low-cost device for creatinine detection. The assembly incorporates metal oxides, which promote electron transfer reactions at low potential, with outstanding electrocatalytic activity, high stability, and low cost, and the incorporation of graphene has unique structural and electrical properties. The paper-based sensor exhibited a linear range from 0.01–2.0 mM and a LOD of 0.22 μM creatinine [173]. Further, Sun et al. designed an electrochemical biosensor that combined MIPs and hybridization chain reaction into microfluidic paper-based analytical devices for ultrasensitive detection of target glycoprotein ovalbumin (OVA) by DPV. The composite of MIPs, including MPBA, captured target glycoprotein OVA. In addition, SiO_2_@Au-nanohybrid-labeled MPBA, and cerium dioxide (CeO_2_)-modified nicked DNA double-strand polymers (SiO_2_@Au/dsDNA/CeO_2_) as a signal tag was captured onto the surface of the electrode in the presence of OVA. As a result, the electrochemical biosensor exhibited a wide linear range of 1 pg/mL to 1000 ng/mL with a LOD of 0.87 pg/mL [174].

It is important to remark that paper-based biosensors have gained attention as they offer a cost-effective and accessible diagnostic alternative for biodetection of clinically relevant molecules/biomarkers. Furthermore, nanobiohybrid materials coupled to paper-based platforms can boost the performance of the resultant systems. In this context, Wang et al. reported a label-free, microfluidic, paper-based, electrochemical aptasensor for ultrasensitive and simultaneous multiplexed detection of CEA and neuron-specific enolase (NSE) by DPV. The paper-based device was fabricated through wax printing and screen-printing, which enabled the functions of sample filtration and sample autoinjection. The paper-electrode was functionalized with amino-modified graphene-Thi-AuNPs and Prussian blue (PB)-PEDOT nanocomposites. The microfluidic aptasensor exhibited a linear range from 0.01 to 500 ng/mL for CEA and 0.05–500 ng/mL for NSE, with a LOD of 2 pg/mL for CEA and 10 pg/mL for NSE, respectively [175]. Cao et al. designed a novel, 3D, paper-based, microfluidic, screen-printed electrode composed of one layer used to print the working electrodes and another layer used to print counter electrodes and reference electrodes assembled by photolithography and screen-printing technology for quantitative detection of glucose by chronoamperometry, as shown in Figure 7IV. PB was deposited on rGO-tetraethylenepentamine (rGO-TEPA/PB) nanocomposite. Then, the GOx enzyme was linked by amide bond formation. Therefore, this 3D, paper-based, microfluidic, electrochemical biosensor responded to glucose over a wide linear range of 0.1 mM–25 mM with a LOD of 25 μM [176]. Schematic illustrations of novel hybrid electrochemical biosensors are shown in Figure 7 and summarized in Table 3.

The significant advance in developing nanobioengineered platforms for electrochemical biosensing has been remarkable in the last five years. However, new 2D and 3D nanomaterials emerge year by year with various improved properties ranging from quantum tunneling, excellent stability, and high conductivity and versatility, which provide new opportunities to develop electrochemical biosensors with high selectivity and extremely low LODs. Furthermore, the appearance of these novel nanostructured materials has led to the implementation of advanced and ultrasensitive biodetection tools.

**Table 3 molecules-27-03841-t003:** Examples of nanobioengineered biosensors, indicating the nanobiohybrid (nanomaterial and biomolecules) and analytic characteristics.

Biosensor	Application ^a^	Nanobiohybrid: Nanomaterial and Biomolecules ^b^	Characterization ^c^	Analytical Performance (Linear Range and LOD) ^d^	Reference
Immunosensor	PSA	Antibody/HP5@AuNPs@g-C_3_N_4_ bioconjugated with PSA-Ab2	CV, EIS, and DPV	0.0005 to 0.00 ng/mL with LOD of 0.12 pg/mL	[117]
	HER2	Ab/g-C_3_N_4_/AuNPs/Cu-MOF	CV and EIS	1.00 to 100.00 ng/mL with LOD of 3.00 fg/mL	[129]
	AXL	Ab/fGQDs	XRD, FTIR, UV-Vis, TEM, EIS, DPV	1.7 to 1000 pg/mL with LOD of 0.5 pg/mL	[130]
	CEA	CdSe-QD-melamine and Ab1-TiO_2_-AuNP-ITO	DPV	0.005–1000 ng/mL with a LOD of 5 pg/mL	[155]
	CA19-9	CeO_2_/FeOx@mC	XPS, TEM, EIS, CV	0.1 mU/mL to 10 U/mL with a LOD of 10 μU/mL	[160]
	NMP-22	Co-MOFs/CuAu NWs/Ab	SEM, XPS, CV, and chronoamperometry	0.1 pg/mL to 1 ng/mL with a LOD of 33 fg/mL	[163]
Genosensor	Zika	Anti-Dig-HRP	Chronoamperometry, CV, EIS	5 to 300 pmol/L with LOD of 0.7 pM	[132]
	Zika genes	AuNPs/ssDNA	SEM, CV, DPV, and chronoamperometry	10 to 600 fM with LOD of 0.2 fM	[133]
	CaMV35S gen	Fe_3_O_4_-Au@Ag-sDNA on MWCNT/AuNPs/SH-sDNA	TEM, XRD, UV-Vis, CV, and DPV	1 × 10^−16^ M to 1 × 10^−10^ M with LOD of 1.26 × 10^−17^ M	[134]
	mi-R21	3-(trimethoxysilyl)propyl methacrylate/ITO/PET/Fc-hybrid DNA hydrogel	DPV	10 nM to 50 μM with a LOD of 5 nM	[157]
	miRNA-122	rGO/Au/DNA	XRD, TEM, Raman, XPS, CV, and DPV	10 μM to 10 pM with a LOD of 1.73 pM	[166]
	OVA	SiO_2_@Au/dsDNA/CeO_2_	DPV	1 pg/mL to 1000 ng/mL with a LOD of 0.87 pg/mL	[174]
Enzymatic	Glucose	GOx/n-TiO_2_/PANI	CV and chronoamperometry	0.02 to 6.0 mM with LOD of 18 μM	[106]
	Glucose	Cu-nanoflowers-Gox-HRP/AuNPs-GO-PVA nanofibers	UV–Vis, SEM, TEM, XDR, CV, and chronoamperometry	0.001 to 0.1 mM with a LOD of 0.018 μM	[158]
	Organophosphate pesticides	acetylcholinesterase/chitosan-transition metals/graphene/GCE	SEM, TEM, XPS, XRD, CV, DPV and EIS	11.31 μM to 22.6 nM with LOD of 14.45 nM	[165]
	β-hydroxybutyric acid	Ti_3_C_2_Tx nanosheets conjugated with β-hydroxybutyrate dehydrogenase	SEM, CV, and chronoamperometry	0.36 to 17.9 mM with a LOD of 45 μM	[168]
Based on peptides	norovirus	Cys/peptide/gold layer	CV and EIS	The LOD was 99.8 nM and 7.8 copies/mL for rP2 and human norovirus, respectively.	[122]
	PSA	MXene-Au-MB nanohybrid/peptide	DPV	5 pg/mL to 10 ng/mL with a LOD of 0.83 pg/mL	[169]
	PKA and CK2	Peptide/MSF/ITO	Chronoamperometry	The LODs were 0.083 and 0.095 U/mL, for PKA and CK2, respectively	[170]
	NHE	Cys-PEG-QRRMIEEPA-MB	DPV and SWV	10 and 150 nM with a LOD of 250 pM	[177]
Based on glycoproteins	*Toxoplasma gondii*	Ab glycosylphosphatidylinositol/SPAuE	CV, EIS	1.0 to 10.0 IU/mL, with a LOD of 0.31 IU/mL	[140]
	MIPs/glycoproteins	Fc/MPBA/AuNPs-SiO_2_ nanobioconjugate	FTIR, CV, EIS, DPV, and chronoamperometry	1 pg/mL to 100 ng/mL and reached a LOD of 0.57 pg/mL	[142]
Based on aptamers	tumor exosomes extracted from lymph node carcinoma of a prostate cells line	MNPs/aptamer-DNA/double-stranded DNA/GCE	DPV	The LOD was 70 particles/μL	[141]
	miRNA	DSN/AuNPS/HRP	CV, EIS and chronoamperometry	The LOD was 43.3 aM	[156]
	CA125 and living MCF-7 cells	Tb-MOF-on-Fe-MOF	SEM, TEM, XPS, CV, and EIS	100 μU/mL to 200 U/mL with a LOD of 58 μU/mL towards CA125. Moreover, biosensor detecting MCF-7 cells with a LOD of 19 cells/mL	[161]
	CEA and NSE	Paper-electrode functionalized with amino-modified graphene-Thi-AuNPs and PB-PEDOT	DPV	0.01 to 500 ng/mL for CEA and 0.05–500 ng/mL for NSE with a LOD of 2 pg/mL for CEA and 10 pg/mL for NSE, respectively	[175]
Other types of biosensors (based on cells or mimicking biosensors)	Impedimetric biosensor/*Escherichia coli B*.	CNT/PEI-T2 virus/GCE	EIS	10^3^ to 10^7^ CFU/mL with LOD of 1.5 × 10^3^ CFU/mL	[135]
	Nonenzymatic biosensor/glucose	GS/GNR/Ni	Chronoamperometry	5 nM to 5 mM with a LOD of 2.5 nM.	[159]
	Mimicking biosensor/H_2_O_2_ released from H9C2 cardiac cells	AuNFs/Fe_3_O_4_@ZIF-8-MoS_2_	SEM, fluorescence, CV, EIS, and chronoamperometry	5 μM–120 mM and a LOD of 0.9 μM	[162]
	Electrochemical/glucose	CuOx@Co_3_O_4_ core-shell nanowires/ZIF-67	SEM, TEM, XRD, XPS, CV, and chronoamperometry	0.1 to 1300.0 μM with a LOD of 36 nM	[164]
	Mimicking/L-tyrosinase	UT-g-C_3_N_4_/Ag hybrids	TEM, XPS, XRD, AFM, EIS, CV, and DPV	1.00 × 10^−6^ to 1.50 × 10^−4^ mol/L with a LOD of 1.40 × 10^−7^ mol/L	[167]
	Biomimetic biosensor/glucose	Fe_3_O_4_@PNE-GOx	Chronoamperometry	0.24 to 24 mM with a LOD of 6.1 µM	[171]
	PAD/creatinine	CuO/IL/ERGO/SPCE	Chronoamperometry	0.01 to 2.0 mM and a LOD of 0.22 μM	[173]
	3D paper-based microfluidic electrochemical biosensor/glucose	rGO-TEPA/PB	SEM, Raman, CV, and chronoamperometry	0.1 mM–25 mM with a LOD of 25 μM	[176]

^a^ PSA, prostate-specific antigen; HER2, human epidermal growth factor receptor 2; AXL, tyrosine kinase; CaMV35S, cauliflower mosaic virus 35S; SAMs, self-assembled monolayers; MIPs, molecular imprinted polymers; miRNA, micro-ribonucleic acid; CA19-9, carbohydrate antigen 19-9; CA125, carbohydrate antigen 125; NMP-22, nuclear matrix protein-22; PKA, protein kinase A; CK2, casein kinase II; PAD, paper-based analytical devices; OVA, ovalbumin; CEA, carcinoembryonic antigen; NHE, human neutrophil elastase; NSE, neuron-specific enolase. ^b^ HP5, hydroxylpillar[5]arene; AuNPs, gold nanoparticles; Ab, antibody; Gox, glucose oxidase; PANI, polyaniline; Cys, cysteine; MOFs, metal–organic framework; fGQDs, functionalized graphene quantum dots; anti-Dig-HRP, antibody-digoxigenin-horseradish peroxidase; ssDNA, single-strand DNA; MWCNT, multiwalled carbon nanotube; CNT, carbon nanotube; PEI, polyethyleneimine; GCE, glassy carbon electrode; SPAuE, screen-printed gold electrode; MNPs, metallic nanoparticles; Fc, ferrocene; MPBA, 4-mercaptophenylboronic acid; QD, quantum dots; ITO, indium tin oxide; DSN, duplex-specific nuclease; PET, polyethylene terephthalate; GO, graphene oxide; PVA, poly(vinyl alcohol); GS, graphene sheet; GNR, graphene–gold nanorod; rGO, reduced graphene oxide; UT, ultrathin; PEG, polyethylene glycol; MB, methylene blue; THI, electron-mediating thionin; PB, Prussian blue; PEDOT, poly(3,4-ethylenedioxythiophene), SPCE, screen-printed carbon electrode; MSF, mesoporous silica thin film; PNE, polynorepinephrine; IL, ionic liquid; ERGO, electrochemically reduced graphene. ^c^ CV, cyclic voltammetry; EIS, electrochemical impedance spectroscopy; DPV, differential pulse voltammetry; XRD, X-ray diffraction; XPS, X-ray photoelectron spectroscopy; FTIR, infrared spectroscopy; UV–Vis, ultraviolet visible spectroscopy; SEM, scanning electron microscopy; TEM, transmission electron microscopy; SWV, square wave voltammetry. ^d^ LOD, limit of detection.

## 7. Limitations, Opportunities, and Concluding Remarks

Biosensor technology based on nanobiohybrid materials represents a vast field that significantly impacts healthcare, the environment, and food quality control. These functional platforms promote target molecule detection with high specificity and sensitivity, particularly in the biomedical field [1,33,50,93,117,178]. Furthermore, the rational design of the nanobiohybrids has been demonstrated to enhance the response and long-term stability of the resultant devices due to the incorporation of nanomaterials with improved properties that promote a favorable nanoenvironment for bioreceptors anchoring. Besides, the versatility of nanomaterials facilitates the conjugation with molecules by multiple conjugation chemistry, opening options to detect numerous target molecules.

Electrochemical-based nanohybrid biosensors have the potential to solve most of the limitations and concerns of bioanalysis and diagnostic tests while maintaining the required sensitivity, selectivity, and LOD to face real needs. Besides, integrating sample preparation into the device allows the possibility of direct analysis within a sample matrix and offers opportunities for new strategies of long-term analysis in vivo, among many other exciting applications. Electrochemical nanohybrid biosensors are particularly suitable for miniaturization and integration in microfluidic devices, thus reducing the consumption of reagents and samples [179,180]. Applications include detecting whole cells, cell components, proteins, and small molecules to address diagnostics and food and environmental control tasks online and in real-time, but still require more sophisticated platforms with additional elements, such as sample preparation. Although nanobioengineered biosensors are an affordable analytical strategy relative to gold standard detection methods, the development of large-scale electrochemical nanobiosensors is still challenging because they require state-of-the-art technologies for their production in a reproducible and stable manner, directly influencing the cost of the sensing device [178,181]. This apparent drawback could be overcome by scaling, automation, and mass manufacturing to lower costs through advanced methods in elaborating cost-affordable and disposable electrochemical nanobiosensors based on additive manufacturing, including screen inkjet 3D printing or microfabrication technologies [44,181,182,183].

Overall, this review exemplified nanobiosensors mainly based on screen-printed electrodes modified with nanohybrids conjugated with highly specific bioreceptors for enhanced biosensing. Yet, the richness in the art of biosensors deserves deeper exploration and support of exciting new ideas. Overall, nanobiohybrids are paving the way in the pioneering development of highly sensitive and selective electrochemical nanobiosensors and represent remarkable research advances that are a step forward in increasing the impact of this exciting, cutting-edge technology in the field of biomarker detection of clinical interest [181].

## Figures and Tables

**Figure 1 molecules-27-03841-f001:**
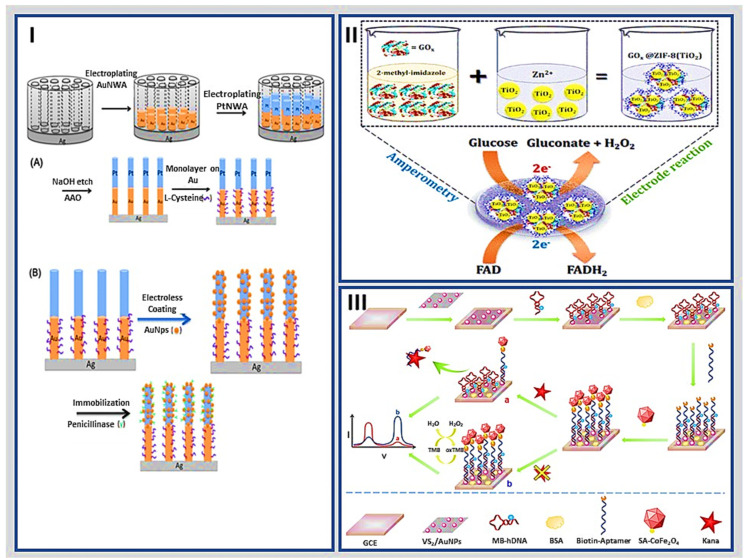
Schematic illustration of metallic-nanomaterial-based biosensors. (**I**) Schematic illustration of the fabrication process of (**A**) Au-Pt multisegmented nanowire array and with L-cysteine immobilization on Au segment; (**B**) electroless plating of Au nanoparticles on Pt segment followed by penicillinase enzyme immobilization, reprinted from Li et al. [50] copyright Elsevier 2019. (**II**) Amperometric Glucose biosensor using MOF-encapsulated TiO_2_ platform, reprinted from Paul et al. [51] copyright Elsevier 2018. (**III**) Schematic illustration of the preparation process of the aptasensor based on VS_2_/AuNPs and the electrochemical detection strategy of the Kana, reprinted from Tian et al., copyright Elsevier 2020 [52].

**Figure 2 molecules-27-03841-f002:**
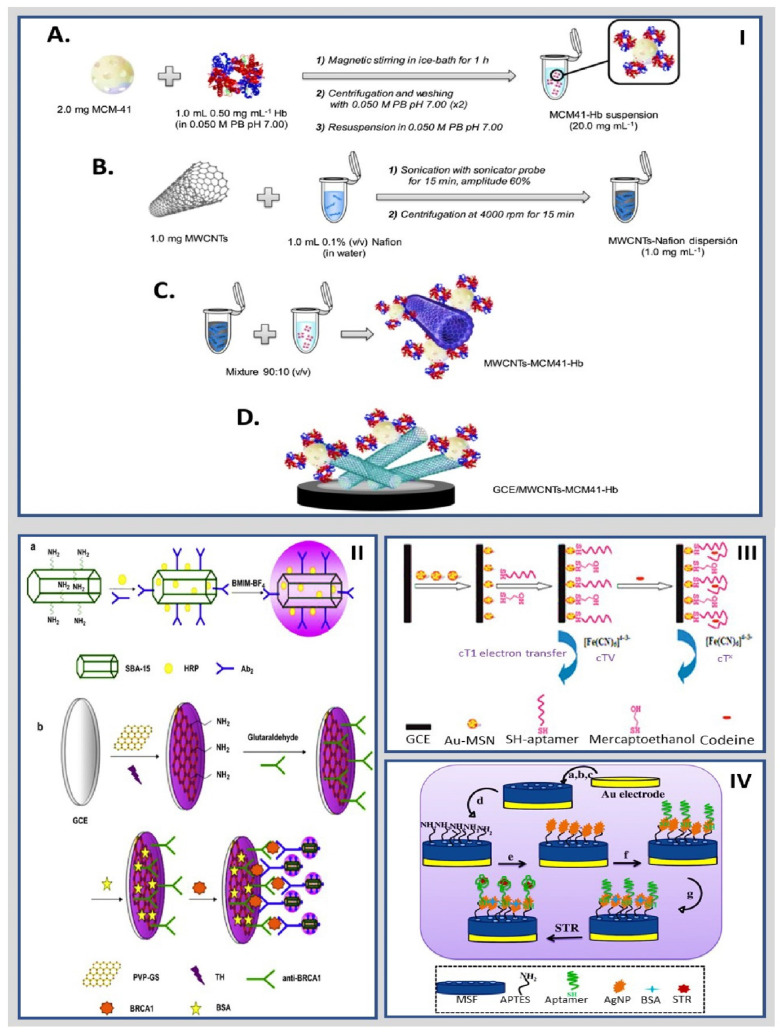
Schematic illustration of silicon-nanomaterial-based electrochemical biosensors. (**I**) schematic representation of the steps involved in preparing the glassy carbon electrode (GCE)/MWCNTs-MCM41-Hb bioplatform, reprinted from Eguílaz et al. [62] copyright Elsevier 2018. (**II**) Electrochemical immunosensor for BRCA1 using BMIM·BF4-coated SBA-15 as labels and functionalized graphene as an enhancer, reprinted from Cai et al. [64] copyright Elsevier 2011. (**III**) The stepwise procedure of label-free electrochemical biosensor based on a DNA aptamer against codeine, reprinted from Huang et al. [65] copyright Elsevier 2013. (**IV**) Schematic representation of the electrochemical aptasensor for streptomycin based on covalent attachment of the aptamer onto a mesoporous silica-thin-film-coated gold electrode, reprinted from Roushani et al. [66] copyright Elsevier 2019.

**Figure 3 molecules-27-03841-f003:**
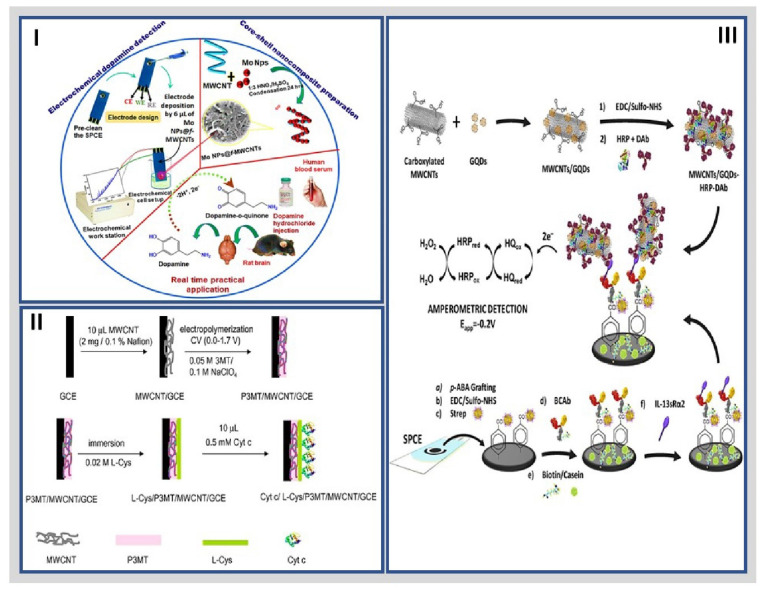
Schematic illustration of CNT hybrid-based electrochemical biosensors. (**I**) Schematic representation for preparing core-shell Mo NPs@f-MWCNTs hybrid nanocomposite and its electrochemical determination of neurotransmitter in biological samples, reprinted from Keerthi et al. [94] copyright Elsevier 2019. (**II**) Scheme showing the stepwise preparation of Cyt c/l-Cys/P3MT/MWCNT/GCE biosensor, reprinted from Eguílaz et al. [95] copyright Elsevier 2010. (**III**) Schematic display of the different steps involved in assembling the amperometric sandwich-like immunosensor for IL-13sRα2 based on the immobilization of BCAb onto Strep/p-ABA/SPCE and the use of MWCNTs/GQDs-HRP-DAb nanocarriers., reprinted from Serafín et al. [96] copyright Elsevier 2019.

**Figure 4 molecules-27-03841-f004:**
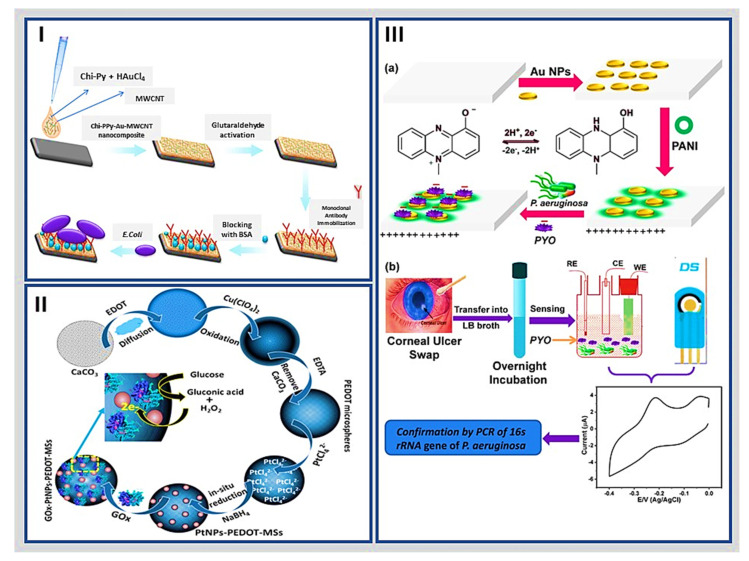
Schematic illustration of polymeric hybrid-based electrochemical biosensors. (**I**) Schematic diagram of the experimental setup of an electrochemical immunosensor using chitosan, MWCNT, polypyrrole with AuNP-hybrid sensing platform for sensitive detection of *Escherichia coli* O157:H7, reprinted from Güner et al. [105] copyright Elsevier 2017. (**II**) Schematic representation of the GOx-PtNPs-PEDOT-MSs biohybrid conducting polymer composite for glucose detection, reprinted from Liu et al. [107] copyright Elsevier 2018. (**III**) (**a**) Schematic diagram for fabrication of PANI/AuNPs/ITO electrode and the interaction between PANI and PYO; (**b**) schematic diagram for electrochemical sensing of *P. aeruginosa* by using either PANI/AuNPs/ITO or SPE electrodes, reprinted from Khalifa et al. [108] copyright Elsevier 2019.

**Figure 5 molecules-27-03841-f005:**
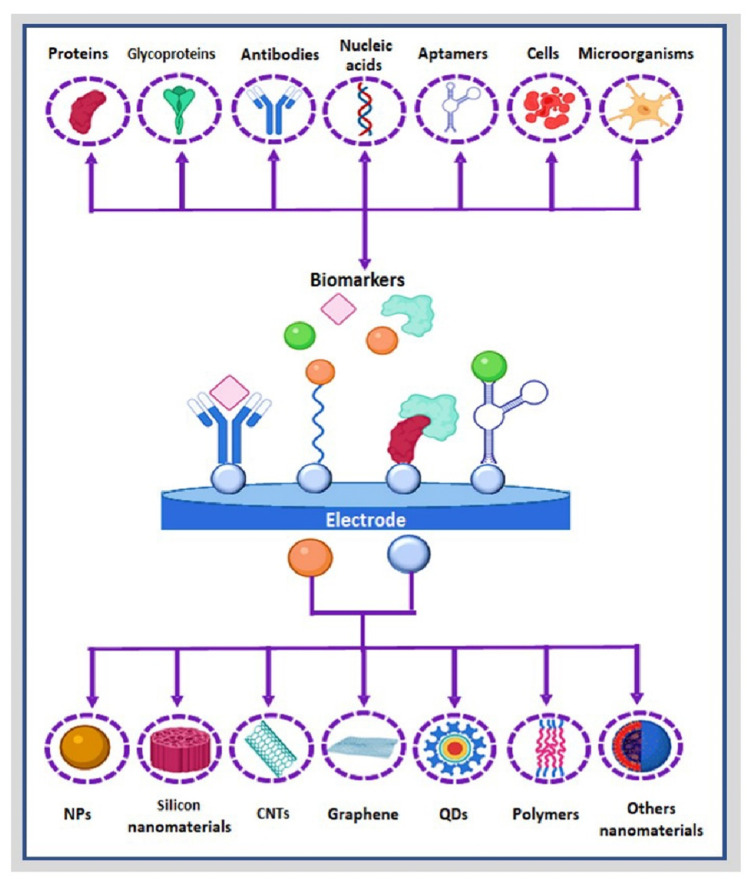
Scheme of electrochemical biosensors based on proteins, glycoproteins, antibodies, nucleic acids, aptamers, cells, or microorganisms at an electrode surface decorated with NPs, silicon nanomaterials, CNTs, graphene, QDs, polymers, or other nanomaterials.

**Figure 6 molecules-27-03841-f006:**
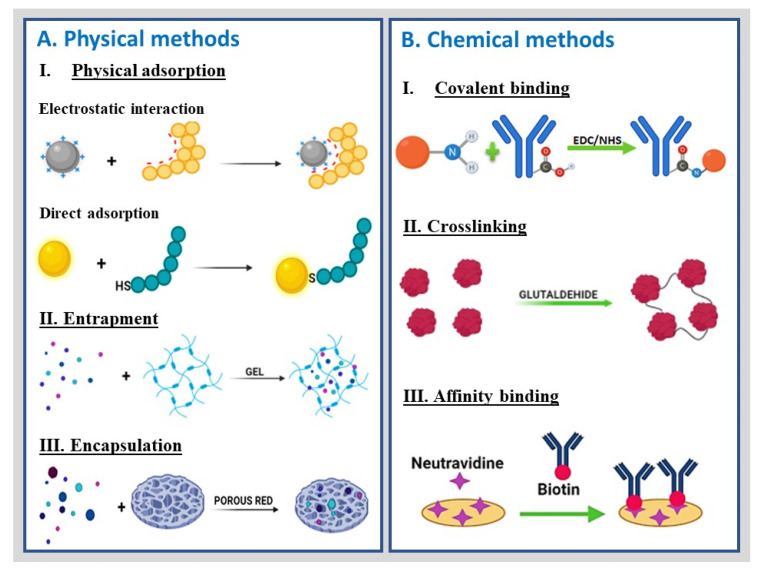
Conjugation of nano(bio)sensors involves binding the bioreceptor’s specific and oriented form to the transducer surface by physical (**A**) or chemical methods (**B**). (**A**) The physical methods include (**I**) physical adsorption; (**II**) enzyme entrapment in a sol–gel, hydrogel, or paste, confined by semipermeable membranes; and (**III**) encapsulation. (**B**) The chemical methods include (**I**) covalent binding, (**II**) crosslinking, and (**III**) affinity binding.

**Figure 7 molecules-27-03841-f007:**
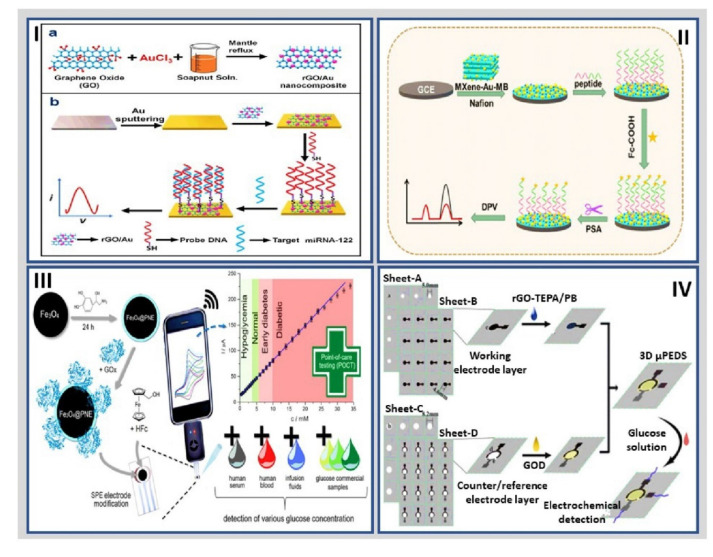
Schematic illustration of novel hybrid electrochemical biosensors. (**I**) Schematic representation of (**a**) synthesis of rGO/Au nanocomposite and (**b**) fabrication of rGO/Au nanocomposite-based miRNA-122 electrochemical detection platform, reprinted from Kasturi et al. [166] copyright Elsevier 2021. (**II**) Schematic representation of the Mxene-Au-MB-peptide biohybrid for PSA detection, reprinted from Xu et al. [169] copyright Elsevier 2021. (**III**) Portable glucose biosensor based on polynorepinephrine@magnetite nanomaterial integrated with a smartphone analyzer for point-of-care application, reprinted from Jędrzak et al. [171] copyright Elsevier 2022. (**IV**) Preparation of 3D, paper-based, microfluidic, electrochemical biosensor, reprinted from Cao et al. [176] copyright Elsevier 2020.

**Table 2 molecules-27-03841-t002:** Characterization techniques of hybrid nanomaterials, nanobioconjugates and electrochemical biosensors.

Techniques	Physicochemical Characteristics Analyzed
Fourier transform infrared spectroscopy (FTIR).	This technique characterizes the functional groups, surface properties, structure, and conformation of hybrid nanomaterials and nanobioconjugates.
Thermogravimetric analysis (TGA).	Thermogravimetric analysis of nanohybrids determines their thermal stability by estimating organic and inorganic material extent.
Ultraviolet spectroscopy (UV-Vis).	This technique can be used to estimate variables such as *K_m_* and *V_max_* in enzyme nanobioconjugates.
Dynamic light scattering (DLS).	This technique can estimate the hydrodynamic size distribution of nanostructures.
Electrophoretic light scattering.	The stability of nanomaterials is highly dependent on the surface charge, among other factors.
X-ray diffraction (XRD).	These techniques characterize hybrid nanomaterials’ size, shape, and crystalline structure.
X-ray photoelectron spectroscopy (XPS).	
Transmission electron microscopy (TEM).	Imaging techniques study size, size distribution, aggregation, dispersion, heterogeneity, morphological characteristics, and compositional analysis of the hybrid nanomaterials and nanobioconjugates.
Scanning electron microscopy (SEM).	
Electrochemical techniques.	Electrochemical techniques such as CV and EIS are used to evaluate electron transfer before, during, and after the bioreceptors attach to the surface of hybrid nanomaterials. They are also used to characterize the analytical properties of the resultant biosensors.

## Data Availability

Not applicable.

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
