# Peer review of "Hybrid Nanobioengineered Nanomaterial-Based Electrochemical Biosensors"

_molecules, 2022, doi:10.3390/molecules27123841_

Round 1

Reviewer 1 Report

This paper reviews nanohybrid and nanocomposite materials and their conjugation to biomolecules for electrochemical biosensing. It also describes recent examples (last 5 years) of nanobioengineered biosensors. The authors discuss the main limitations and perspectives regarding this topic. This review is quite complete and relevant to researchers in the field of biosensors. Nevertheless, it does not include one of the most intensive research topics in carbon based nanomaterials that have emerged in the last years: carbon dots-nanohybrids. The following suggestions should be also addressed:

- Table 2. The table does not properly show which rows in the first column correspond to the combined rows in the second column.

- Line 576: It is not clear the idea that the authors aim at communicating in this sentence

- Lines  591 and 760: What do GE and MB stand for?

- Table 3: It would be clearer for readers to structure the examples list according to a criteria for classification, such as type of biosensor.

Author Response

We thank reviewer 1 for recognizing that "the review is quite complete and relevant to researchers in the field of biosensors" and for recommending it for publication after revisions.

Reviewer 2 Report

This paper reviews the development of nanobioengineered nanomaterial for applications in electrochemical biosensors. Various types of engineered nanomaterials are summarized and their structural properties and features for incorporation into biosensing systems are discussed. Lastly, the limitations and opportunities of these hybrid nanomaterials in enhancing electrochemical platforms are presented. The sectioning in this review is confusing and the importance/need for this review is not presented clearly. There are some comments that need to be addressed:

  1. Graphical abstract: Provided, the inner circle is a sandwich detection platform but which is not often the case used in electrochemical biosensor. In addition, it would be clearer if the author could add in some descriptions for each of the illustrations presented.
  2. Highlight: Not provided, the author should present the highlight and novelty of this review in point form.
  3. An abstract is often presented separately from the article, so it must be able to stand alone. Hence the problem statement, aim, novelty and importance of the study have to be all included in.
  4. Introduction: The author should identify the current gap in the available reviews and how does this review fit into filling the gap. Also, the author did not highlight the importance of applying nanobioengineered material in electrochemical biosensing system.
  5. Introduction: The author could elaborate on the application of nanomaterials in electrochemical biosensing to provide a more comprehensive study background. Kindly refer to relevant papers:
  6. “Recent progress in nanomaterials modified electrochemical biosensors for the detection of MicroRNA”
  7. “Engineered nanomaterial assisted signal-amplification strategies for enhancing analytical performance of electrochemical biosensors”
  8. Section 2: The full stop should be removed from the section title.
  9. Figures: The figures need to be supplied in higher resolution and make sure to retain the proper figure length:width ratio.
  10. Table 1: The table is too simple with too little information. I believe much works have been done within the last ten years and the author should put in more effort in comparing the different works done.
  11. Table 1: The author should provide a footnote to explain all the abbreviation used in Table 1.
  12. Section 3: The author should point out the conjugation method/approach in generating nanohybrid materials rather than just summarizing the works done on nanomaterial-modified electrochemical biosensor. The difference between Section 2 and 3 is not clear as both sections are discussing on the nanomaterial-modified electrochemical biosensor.
  13. Figure 5: The author needs to explain the interaction/conjugation that are presented on the electrode.
  14. Figure 6B: I. should be Covalent binding.
  15. Section 4: The author could provide comparison on the different methods used in conjugation, like the advantages and disadvantages of each method.
  16. The author needs to explain the abbreviation in full form for its first appearance in text, eg. CV, EIS, FTIR, DPV, SEM, DLS and more.
  17. Line 652: What is meant by “DVP”?
  18. How is Section 6 different from the previous sections? What is the significance in highlighting the works done in the last 5 years but not describing on the advancement in the field.

Author Response

We thank reviewer 2 for his comments, addressed the issues, and provided some clarification needed.

Reviewer 3 Report

The manuscript entitled “Hybrid nanobioengineered nanomaterial-based electrochemical biosensors” is an interesting study which classifies some recently engineered organic/inorganic metallic-, silicon-, carbonaceous- and polymeric-nanomaterials and describes their structural properties when incorporated into biosensing systems. It shows the latest advances in ultrasensitive electrochemical biosensors engineered, exemplifying nanobioengineered platforms for electrochemical biosensing modified with hybrid or nanocomposite materials.

The paper is generally well written and structured. The manuscript has a potential to be accepted, after the authors have addressed the following comments and questions:

P1, L10: “Nanoengineering biosensors for monitoring target analytes has become more precise and 10 sophisticated, …”, a correction needs: …monitoring target analytes have become…

P2, L48, last paragraph: please check if the cited reference (Ref [7]), that it has been address, it is the right one!

P5, Figure 1: the label III is missing in the figure 1!

P6, L203: “…Hb-functionalized mesoporous MCM-41 silica for nitrite trichloroacetic acid, followed…”, and is missing between nitrite and trichloroacetic acid.

P6, L206: “…rapid response to the changes in NO2- and trichloroacetic acid…”, formula of nitrite needs to modify to NO2-

P6, L219: “…immunoassay exhibited a wide lineal range from…”, a typographic correction needs …linear…

P12, L417: “…from 238 μM to 1.9 μM and …”, it should be reported from low to high concentration i.e., from 1.9 μM to 238 μM…, it is also suggested to cite the origin paper i.e., Elkhawaga A.A., Khalifa M.M., El-badawy O., Hassan M.A., El-Said W.A. (2019) Rapid and highly sensitive detection of pyocyanin biomarker in different Pseudomonas aeruginosa infections using gold nanoparticles modified sensor. PLoS ONE 14 (7): e0216438. https://doi.org/10.1371/journal. pone.0216438

P12, caption of Figure 4: “Schematic illustration of CNT hybrid-based electrochemical biosensors”. This figure is illustrating schematic of Polymeric hybrid-based electrochemical biosensors. 

P8, L584-L587: ”Synthetic biomimetic receptors are designed and fabricated to mimic an actual bioreceptor (antibody, enzyme, cell or nucleic acids) to simulate the binding target molecules with similar affinities and specificities to natural receptors but much more long-term robustness and stability (Figure 5). “

The authors are describing the definition of molecularly imprinted polymers (Synthetic biomimetic), and they have addressed the figure 5, however this figure is illustrating the scheme of electrochemical biosensors not synthetic biomimetic.

P24, L735: “…a shell that calcinated from using microporous ZIF-67. …”, ...from … is enough!

P24, L756: “The biosensor displayed a sensitivity of 0.480 μA/Mm, …”the unit of sensitivity needs to modify to 0.480 μA mM-1 cm-2

P 32, References:

Ref. [2]: Nic, M.; Jirat, J.; Kosata, B. Definitions of terms relating to the structure and processing of sols, gels, 894 networks, and inorganic-organic hybrid materials (IUPAC Recommendations 2007). In IUPAC. 895 Compendium of Chemical Terminology; 2007; p. 1801.

Please check if the names of authors are correct.

Please check the style of addressed references below:

Ref. [5]: Meral Yüce; Hasan Kurt How to make nanobiosensors: surface modification and characterisation of nanomaterials for biosensing applications. RSC Adv. 2017, 7, 49386–49403.

Ref [8]: Yanling Hu; Ying Huang; Chaoliang Tan; Xiao Zhang; Qipeng Lu; Melinda Sindoro; Xiao Huang; Wei Huang; Lianhui Wang; Hua Zhang Two-dimensional transition metal dichalcogenide nanomaterials for biosensing applications. Mater. Chem. Front. 2016, 1, 24–36.

Ref. [75]: Zhao, C.; Song, X.; Liu, Y.; Fu, Y.; Ye, L.; Wang, N.; Wang, F.; Li, L.; Mohammadniaei, M.; Zhang, M.; et 1068 al. Synthesis of graphene quantum dots and their applications in drug delivery. J. Nanobiotechnology 2020 1069 181 2020, 18, 1–32.

Ref. [115]: Janeway CA Jr, Travers P, Walport M, et al. The structure of a typical antibody molecule. In 1171 Immunobiology: The Immune System in Health and Disease.; Science, N.Y.G., Ed.; New York: Garland 1172 Science, 2001.

Author Response

We thank reviewer 3 for his comments and comprehensive review of the document.
